# Tyrosine residues initiated photopolymerization in living organisms

Mei Zhu[1], Shengliang Wang[1], Zhenhui Li[1], Junbo Li[1], Zhijun Xu[1], Xiaoman Liu [1] ✉ & Xin Huang [1] ✉

Towards intracellular engineering of living organisms, the development of new biocompatible polymerization system applicable for an intrinsically non-natural macromolecules synthesis for modulating living organism function/behavior is a key step. Herein, we find that the tyrosine residues in the cofactor-free proteins can be employed to mediate controlled radical polymerization under 405 nm light. A proton-coupled electron transfer (PCET) mechanism between the excited-state TyrOH* residue in proteins and the monomer or the chain transfer agent is confirmed. By using Tyr-containing proteins, a wide range of well-defined polymers are successfully generated. Especially, the developed photopolymerization system shows good biocompatibility, which can achieve in-situ extracellular polymerization from the surface of yeast cells for agglutination/anti-agglutination functional manipulation or intracellular polymerization inside yeast cells, respectively. Besides providing a universal aqueous photopolymerization system, this study should contribute a new way to generate various non-natural polymers in vitro or in vivo to engineer living organism functions and behaviours.

Owing to its high compatibility and versatility, enzyme-mediated polymerization has been promising for the development of intra-organism polymerization[1-3], living material synthesis as well as disease therapies[4-7]. Over the past decade, there has been a significant effort towards developing protein-mediated polymerization strategies with an alkyl bromide initiator by atom transfer radical polymerization (ATRP), while horseradish peroxidase[8,9], hemoglobin[10] and laccase[11] have been successfully confirmed to achieve it. Moreover, recently An and co-workers reported that cofactor-proteins such as flavoproteins glucose oxidase (GOx) and pyranose 2-oxidase (P₂Ox) showed the unnatural photoenzymatic function to catalyze oxidation of β-d-glucose to generate FADH⁻, subsequently facilitating photoinduced electron transfer (PET) process to initiate reversible addition–fragmentation chain transfer (RAFT) polymerization under visible light irradiation[12]. Therefore, to explore the application of enzymes in controlled/living radical polymerizations (CRP), considerable efforts have been focused on cofactor-proteins such as metalloenzymes and flavoproteins. While, the majority of these

polymerization is dependent on specific catalytic process, in which the active center would exhaust substrates or sacrificial reagents[13-17]. In this regard, it will be exciting to further develop a simple and universal method to initiate polymerization with cofactor-free protein, which would definitely boost the development of non-natural polymers synthesizing in living organisms towards modulating organismic functionality/behavior or cellular therapy[18-22].

Photopolymerization has proved to be one of the effective tools for intracellular or extracellular synthesis of non-natural macromolecules in the application of engineered living materials[23-27], and a wide range of organic and inorganic photocatalysts have been reported including Ir/Ru-based photoredox catalysts[28-30], zinc tetraphenylporphyrin[31], dyes[32,33], various inorganic semiconducting nanoparticles[34-37], and magnetic nanoparticles etc[38]. Moreover, Xu and Boyer reported that chlorophyll a could activate a photoinduced electron transfer (PET) process and initiated RAFT polymerization under blue and red LED light[39]. Inspired by these pioneering studies, we try to investigate the possibility to initiate polymerization by

[1]MIIT Key Laboratory of Critical Materials Technology for New Energy Conversion and Storage, School of Chemistry and Chemical Engineering, Harbin Institute of Technology, Harbin 150001, P. R. China. ✉e-mail: liuxiaoman@hit.edu.cn; xinhuang@hit.edu.cn

directly employing proteins which are abundant in living organisms. It is well recognized that aromatic amino acid residues in proteins, such as tyrosine residues (Tyr), have high potential as one-electron redox mediators, while they are generally buried inside proteins, leading to low intrinsic oxidation signals and inefficient determination[40,41]. However, it has been approved that oxidation of Tyr was quite common by proton-coupled electron transfer (PCET) mechanism, resulting in the formation of a neutral radical species[42–44]. In particular, PCET reactions of Tyr are essential to the biological processes derived from ribonucleotide reductase[45,46], photosystem II[47,48] and cytochrome c oxidase[49]. In view of the above observations, in this study, we present a photo-polymerization system initiated by diverse tyrosine-containing cofactor-free proteins. We demonstrate that the excited state of the cofactor-free protein could promote its electron transfer ability, due to the fact that the photo-excited electron donor (or acceptor) shows stronger redox ability[50,51], thereby facilitating its exploitable radical polymerization under visible light, and therefore, the scope of our method can be expanded to the synthesis of non-natural macro-molecules in vitro and in vivo with functional modulation of living organisms (Fig. 1).

## Results

### Tyrosine-containing cofactor-free protein mediated RAFT photopolymerization

To assess the possibility of photopolymerization, different monomers such as 2-methoxyethyl acrylate (**1**) and N-isopropyl acrylamide (**2**) were utilized as models, and the polymerization was initially performed by mixing bovine serum albumin (BSA) with 10% w/v monomers in aqueous solution under 405 nm LED light and anaerobic condition. As anticipated, the BSA-mediated radical photopolymerization was successfully implemented, while the control experiments indicated that no polymerization was observed in the absence of either BSA or light (Supplementary Figs. 1–3, and Supplementary Table 1). Moreover, a chain transfer agent PEG-CTA (**6**) with good water solubility was synthesized by coupling PEG-amine with N-hydroxysuccinimide 2-(dodecylthiocarbonothioylthio) isobutyrate with estimated molecular weight (Mw) of 2346 g mol⁻¹ (Supplementary Figs. 4–7, and Supplementary Table 2). Then the successful RAFT

polymerization in the presence of both BSA and **6** under the light irradiation was achieved with monomer conversion ratio of 83% (Supplementary Fig. 8, and Supplementary Table 3). To assess the kinetics of BSA-mediated RAFT photopolymerization, polymerization of **3** was carried out in the presence of BSA, **6** and **3** (10% w/v) (BSA:**6**:**3** = 0.144:1:460, molar ratio). After 5.5 h of illumination, the reaction mixture became viscous and >80% monomer conversion ratio was confirmed by ¹H NMR (Fig. 2a, Supplementary Fig. 9). The kinetic plot of $\ln([M]_0/[M_t])$ revealed analogous linear relationship versus time, indicating the constant radical concentration and pseudo-first-order polymerization kinetics (Fig. 2b). Significantly, the molecular weight dispersity of polymers was gradually narrowed with the poly-merization proceeding, and maintained a low polydispersity even at high monomer conversion ratio, which indicated that BSA-mediated RAFT photopolymerization presented a well-controlled process (Fig. 2c, d). Furthermore, the RAFT heterogeneous photopolymeriza-tion of **1** was performed to produce polymer nanoparticles (BSA:**6**:**1** = 0.144:1:350). The poly(**1**) nanoparticles with different diameters over the range between 16 and 80 nm were synthesized by controlling the polymerization time, and the diameter of nanoparticles in H₂O was determined by DLS measurement as shown in Fig. 2e. The inset in Fig. 2e is the corresponding SEM image of poly(**1**) nanoparticles after 12.8 h polymerization. Although the insoluble polymers were gradually produced in aqueous solution, the GPC plots of polymers still main-tained good unimodality and symmetry (Fig. 2f). By altering the con-centration of BSA, it was found that RAFT photopolymerization of **1** was nearly not carried out when the concentration of BSA was lower than 0.005 mM (Supplementary Fig. 10, and Supplementary Table 4). Alternatively, by enhancing light intensity, a faster polymerization rate was observed (Supplementary Fig. 11, and Supplementary Table 5). In addition, various monomers capable of BSA-mediated RAFT photo-polymerization were investigated, including monomers N-isopropyl acrylamide (**2**), 4-acryloylmorpholine (**4**) and poly (ethylene glycol) methyl ether acrylate (**5**). The homopolymers with a wide range of molecular weight were synthesized, with narrow dispersity and unimodal GPC traces, which well demonstrated the versatility of such studied protein-based photopolymerization strategy (Supplemen-tary Fig. 12).

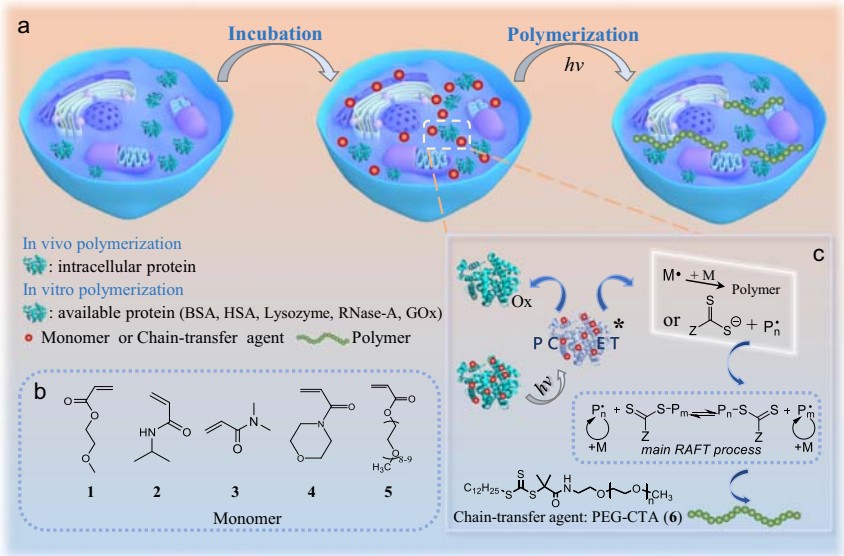

**Fig. 1 | A schematic representation of tyrosine-containing cofactor-free protein mediated radical photopolymerization in vitro and in vivo. a** General process of radical photopolymerization inside yeast cells. **b** Chemical structure of monomers. **c** RAFT radical photopolymerization mediated by tyrosine-containing cofactor-free protein in the presence of a chain-transfer agent.

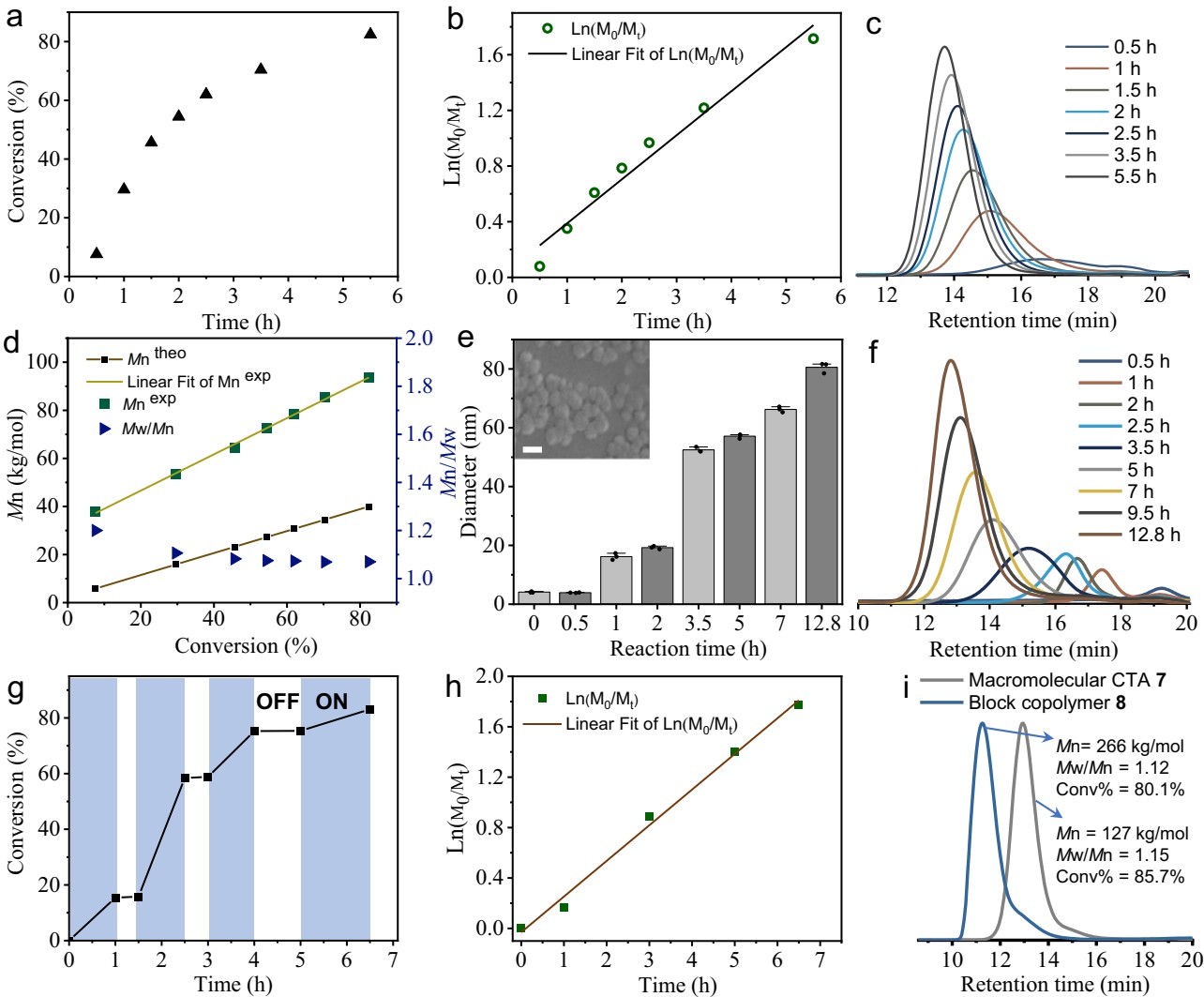

**Fig. 2 | Kinetic study of BSA-mediated RAFT photopolymerization. a** Plot of conversion ratio of **3** versus time. **b** Pseudo-first-order kinetic plot of ln([M]₀/[M$_t$]) versus time. **c** GPC traces of poly(**3**) synthesized at different polymerization time. **d** Evolution of molecular weight and dispersity of poly(**3**) versus monomer conversion ratio. **e** The diameter statistics of the formed poly(**1**) nanoparticles based on DLS measurement in H$_2$O and **f** GPC traces of poly(**1**) during BSA-mediated RAFT photopolymerization of **1** at different time. (The inset in (**e**) is the SEM image of poly(**1**) nanoparticles after 12.8 h polymerization, Scale bar, 100 nm). Data are presented as mean ± s.d., (*n* = 3). **g** Plot of conversion ratio versus time designated ON (blue) / OFF (white) profile and **h** Pseudo-first-order kinetic plot of ln([M]₀/[M$_t$]) versus time in the ON/OFF experiments of BSA-mediated RAFT photopolymerization of **3** in the presence or absence of light. **i** GPC traces of macromolecular CTA **7** and the synthesized block copolymer **8**. Molecular weight (*M*n) and polydispersity (*M*w/*M*n) were determined by GPC analysis (DMF used as eluent). Conversion ratio was determined by $^1$H NMR. Polymerization conditions: [BSA] = 0.315 mM, [M] = 10% w/v, (**a–h**) [BSA]:[**6**]:[**3**] = 0.144:1:460, [BSA]:[**6**]:[**1**] = 0.144:1:350, (**a–d**, **g**, **h**) using **3** monomer and (**e**, **f**) using **1** monomer, (**i**) synthesis of **7**, [BSA]:[**6**]:[**3**] = 0.162:1:517, 6 h, synthesis of **8**, [BSA]:[**7**]:[**3**] = 0.384:1:1230, 6 h, violet LED light (20 W, 405 nm), room temperature, Ar protection.

Spatiotemporal control and few side reactions, are the key characteristics of photopolymerization, which is also applicable for this studied system. To demonstrate so, the polymerization kinetic experiment of BSA-mediated RAFT photopolymerization of **3** (BSA:**6**:**3** = 0.144:1:460) was performed through alternative sequence of light (ON/OFF). In the absence of light, the polymerization was suspended, while triggered immediately in the presence of irradiation. The rapid reversible termination and initiation along with the light switch were summarized in Fig. 2g. The incremental molecular weight of the polymers and the low molecular weight distribution were observed versus time (Supplementary Fig. 13), and as shown in Fig. 2h, the polymerization system still maintained pseudo-first-order polymerization kinetics displaying characteristics of living polymerization with constant radical concentration despite undergoing ON/OFF procedure. Meanwhile, taking the first block polymer **7** as a macromolecular CTA which was prepared using a molar ratio (BSA:**6**:**3** = 0.162:1:517) under the 405 nm

light in H$_2$O for 6 h (with monomer conversion ratio of 85.7%), by mixing with BSA and **3**, the block copolymer **8** was obtained after photopolymerization for 6 h (BSA:**7**:**3** = 0.384:1:1230, monomer conversion ratio of 80.1%). The $^1$H NMR and GPC traces demonstrated the successful synthesis of **8** (Fig. 2i, and Supplementary Fig. 14). Although the conversion of monomer was as high as 80.1%, the block copolymer still maintained single peak and narrow polydispersity. All of the aforementioned observations demonstrated that the polymers synthesized by BSA-mediated RAFT photopolymerization could remain high terminal fidelity.

### Investigation of photopolymerization mechanism

To gain insight into the polymerization mechanism, the change of BSA content during the polymerization was firstly investigated. Based on bradford assay, the concentration of BSA in the supernatant solution of BSA-mediated photopolymerization of **1** at different time was

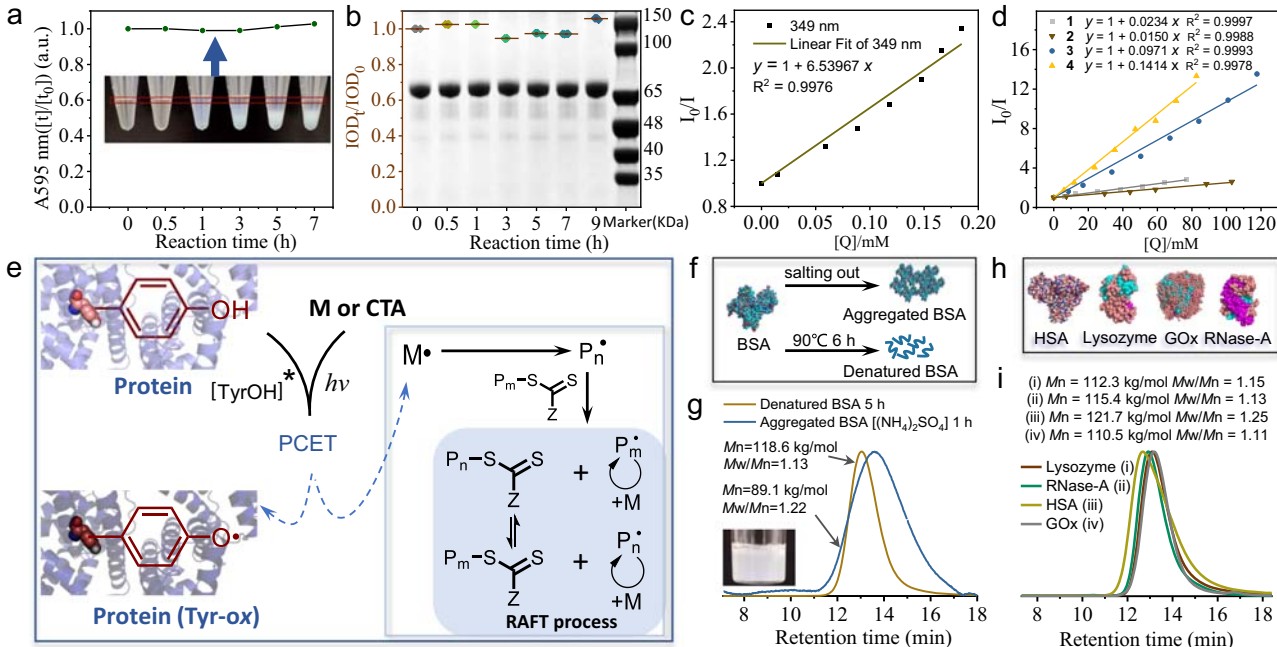

**Fig. 3 | Investigation of BSA-mediated photopolymerization mechanism.**
**a** Determination of BSA concentration in the supernatant during BSA-mediated photopolymerization of **1** at different time by monitoring UV-Vis absorbance at 595 nm due to the combination of coomassie brilliant blue G-250 to proteins. The inset was a photograph of centrifugation after RAFT photopolymerization of **1** at different time. **b** SDS-PAGE and integrated optical density (IOD) of BSA in the supernatant after BSA-mediated RAFT photopolymerization of **1** at different time. Data are presented as mean ± s.d., ($n = 4$). **c, d** Fluorescence quenching studies of BSA quenched with varying concentrations of chain-transfer agent **6** or monomers in $H_2O$. Stern–Volmer plots of the ratio $I_0/I$ versus quencher concentration, (**c**) chain-transfer agent **6** and (**d**) various monomers (**1, 2, 3, 4**). $I_0$ and I correspond to the emission intensity in the absence and presence of quencher, respectively. **e** Proposed mechanism of BSA-mediated RAFT radical photopolymerization. Excited-state BSA under light irradiation could enhance its reduction ability significantly, and then excited-state Tyr residues in BSA interacted with monomers or

CTA, through proton-coupled electron transfer resulting in the formation of primary radical, which subsequently initiated chain propagation or RAFT process. **f** Schematic illustration of BSA structures after salting out (in saturated $(NH_4)_2SO_4$) and denaturation treatment (90 °C, 6 h), respectively. **g** GPC traces of RAFT photopolymerization of **3**, mediated by the aggregated BSA in saturated $(NH_4)_2SO_4$ solution or the denatured BSA in $H_2O$ at room temperature, respectively. The inset is a photograph of the aggregated BSA in saturated $(NH_4)_2SO_4$. **h** The three-dimensional structures of HSA, Lysozyme, RNase-A and GOx. **i** GPC traces of RAFT photopolymerization of **3**, facilitated by four proteins (HSA, Lysozyme, RNase-A, GOx) for 5 h, respectively. Molecular weight ($Mn$) and polydispersity ($Mw/Mn$) were determined by GPC analysis (DMF used as eluent). Conversion ratio was determined by [1]H NMR. Polymerization conditions: [M] = 10% w/v, [BSA] = 0.268 mM, [HSA] = 0.125 mM, [GOx] = 0.048 mM, [Lysozyme] = 0.505 mM, [RNase-A] = 0.608 mM, [**6**] = 1.824 mM, violet LED light (20 W, 405 nm), room temperature, Ar protection.

determined by assessing the variation ratio of the absorbance of coomassie brilliant blue G-250 at 595 nm (Fig. 3a) (BSA:**6:1** = 0.147:1:470). There was no obvious decrease on the concentration of BSA in the supernatant solution after 7 h polymerization in comparison with the original solution. In addition, the SDS-PAGE of supernatant protein demonstrated that the molecular weight of BSA was unchanged. The integrated optical density (IOD) analysis of the gel also proved that the content of BSA in supernatant solution kept constant (Fig. 3b). These results revealed that as a photoredox initiator BSA, did not participate in the formation of protein-polymer bioconjugates. In order to further confirm so, we specially synthesized BSA-poly(**2**) bioconjugates as a control. In comparison with the product obtained by BSA mediated RAFT photopolymerization of **2**, the SDS-PAGE exhibited that the molecular weight of the reaction solution after 6 h (Supplementary Fig. 15, Lane 2) was similar to the original solution (Supplementary Fig. 15, Lane 1) and the mixture of BSA and **2** (Supplementary Fig. 15, Lane 5). However, the lane of BSA-poly(**2**) bioconjugates exhibited the characteristics of increased molecular weight and tailing caused by the conjugated polymers (Supplementary Fig. 15, Lane 3).

To further explore the polymerization mechanism, the electron transfer between BSA and **6** was then investigated. It is recognized that Tyr is one of the main reaction sites during the electrolytic oxidation or photooxidation process of BSA[52–54], and the similar photocurrent responses of BSA and Tyr were also observed under 405 nm light (Supplementary Fig. 19a). As shown in UV-Vis absorption spectrum (Supplementary Fig. 16), Tyr is the major repeated unit in BSA (20 per

BSA) that contributes the UV-Vis absorbance at 278 nm. Therefore, under the 405 nm light irradiation, the degree of excited-state of BSA is closely related with amino acid Tyr. Based on Rehm-Weller equation[51,55], the excited-state oxidation potential of Tyr was estimated to be −3.38 V ($Eox$*(Tyr·+/Tyr*) vs NHE) (Supplementary Fig. 17 and Supplementary Table 6). While in comparison with that of **6** (−0.624 V vs NHE based on cyclic voltammogram, Supplementary Fig. 18), the significantly decreased oxidation potential in photoexcitation indicated that the excited-state BSA could supply sufficient driven force to transfer the electron to **6**. This was further confirmed by a control experiment that Tyr residues in BSA were blocked by forming acetylated BSA and iodinated BSA, respectively, which was then used to mediate the photopolymerization of **1**, respectively, showing lower conversion compared with control BSA (Supplementary Fig. 19b, c)[56]. While due to the oxidation of Tyr, a decrease in the Tyr number in BSA was observed after 9 h polymerization (Supplementary Fig. 19d). Meanwhile, considering that there are 20 Tyr in each BSA, three-cycle polymerization mediated by BSA was also successfully carried out. (Supplementary Fig. 20). Moreover, the direct interaction between the fluorophore Tyr residues in the protein and **6** or **3** was investigated by fluorescence quenching experiment of Ribonuclease A (RNase-A) without containing tryptophan residues (Supplementary Fig. 21). The maximum emission wavelength of RNase-A was 312 nm ($\lambda_{ex}$ = 280 nm) mainly contributed by Tyr. Along with the increasing of the **6** or **3** concentration, the obvious decrease of the emission intensity and red shifts were clearly observed, which then resulted in excited state ionization of Tyr and TyrO· radical

formation[57,58]. Such observation was also consistent with the previous report that excited-state ionization of Tyr can occur even at neutral pH, in which the proton was transferred from excited-state Tyr to proton acceptors, and the reduced emission intensity and the maximum emission wavelength red-shifted of Tyr were exhibited with the increase of the acceptor concentration[59].

Such electron transfer between BSA and **6** or monomers was also confirmed by the fluorescence quenching experiments based on Stern–Volmer plot[29,58]. As shown in Fig. 3c, d, with the addition of **6** or various monomers, the gradual decrease of BSA fluorescence intensity was observed. In particular, the linear relationship between the ratio $I_0/I$ and quencher concentration, well suggested the fluorescence quenching caused by electron transfer between BSA and PEG-CTA or monomers. The parameters of Stern–Volmer equation were summarized in Supplementary Table 7, which showed that **6** had the maximum quencher rate coefficient (Kq, $6.53967*10^{11}$ L mol$^{-1}$ s$^{-1}$) compared with that of the monomers, further indicating the priority of electron transfer between BSA and **6**. To further demonstrate the relationship between quenching ability and polymerization rate, iso-butyl methacrylate and ethyl methacrylate with similar propagation rate coefficient (6.8 and 6.7 °C/$10^{-2}$ L mol$^{-1}$ s$^{-1}$, respectively)[60] were chosen. By quenching experiment, the former exhibited a larger quencher rate coefficient than that of the latter ($0.2885*10^{10}$ and $0.1277*10^{10}$ L mol$^{-1}$ s$^{-1}$, respectively). The RAFT photopolymerization of the two monomers were carried out (BSA:**6**:monomer = 0.339:1:493), and both the plot of ln($[M]_0/[M_t]$) of the two monomers revealed a similar linear relationship versus time while isobutyl methacrylate presented a faster polymerization rate compared with that of ethyl methacrylate, which also indicated the faster quenching rate could correspond to higher polymerization rate (Supplementary Fig. 22). In addition, UV-vis absorbance of the protein and monomers indicated that there was no interaction between monomers and the protein in the form of ground-state complexes (Supplementary Fig. 23).

Moreover, electron spin resonance (ESR) experiment was also employed to confirm the formation of radicals. 5,5-dimethyl-1-pyrroline N-oxide (DMPO) as the spin trapping was mixed with BSA and **3** in H$_2$O (BSA:**3**:DMPO = 0.3:1000:100). The signal of a relatively stable adduct radical ($\alpha_N = 15.9$ G, $\alpha_{\beta\text{-H}} = 21.1$ G) in the ESR spectra clearly revealed radical generation during the photopolymerization (Supplementary Fig. 24). To assess the reduction and the approximate amount of transferred electrons of BSA, 2-phenyl-4,4,5,5-tetramethylimidazoline-1-oxyl-3-oxide (PTIO), a water-soluble free radical compound, was employed. Its charged group is an oxygen radical and the degree of the reaction is quantitatively related to the number of electrons received[61]. The BSA with different concentrations were mixed with PTIO under 405 nm illumination in anaerobic condition. By monitoring the absorbance of PTIO at 557 nm at different time, we could estimate that each BSA transferred more than two electrons to PTIO on average in 10.6 h (Supplementary Fig. 25). Therefore, based on above observations, we proposed the following polymerization process as shown in Fig. 3e. Upon excitation with violet light (405 nm), excited-state TyrOH* residue in BSA transferred the electron to CTA or monomers, followed by proton transfer via PCET mechanism along with the formation of oxidized-state Tyr-ox, and then initial radicals were generated to initiate polymer chain propagation or RAFT process.

To be of great interest, it was found that the excited-state TyrOH* residue in BSA and the electron transfer were independent of the three-dimensional structure of protein. In this regard, we disturbed the structure of BSA by salting out or heating to obtain aggregated BSA or denatured BSA (Fig. 3f), which were subsequently utilized for photo-polymerization, respectively. The RAFT photopolymerization was performed in saturated (NH$_4$)$_2$SO$_4$ solution to maintain the structure of the aggregated BSA with white turbid solution, followed by adding **6** and **3** to the saturated (NH$_4$)$_2$SO$_4$ solution. Although the reaction was carried out in opaque solution, the yellow viscous product was observed after

polymerization for 1 h, and the GPC curve of the polymer also showed obvious unimodal peaks with low molecular weight distribution. Alternatively, BSA was heated to 90 °C for 6 h, hence its three-dimensional structure including secondary structure and tertiary structure was destroyed in order to obtain the extended polypeptides (denatured BSA). Analogously, **6** and **3** were incorporated into the denatured BSA aqueous solution under 405 nm LED irradiation, and the poly(**3**) was also successfully synthesized with a low polydispersity (Fig. 3g).

Inspired by the above investigations that BSA-mediated photo-polymerization was not related to its three-dimensional structure, we envisaged that other Tyr residues containing proteins should also have the potential to initiate the RAFT photopolymerization. Four proteins with different three-dimensional structures, molecular weight and iso-electric points, including cofactor-free proteins (human serum albumin (HSA), Lysozyme, RNase-A) and the cofactor-protein GOx containing 18, 6, 6, and 27 Tyr residues per protein, respectively, were employed to assess the feasibility of polymerization (Fig. 3h and Supplementary Table 8). The RAFT photopolymerization of **3** was performed by utilizing the four different proteins, and successful polymerization was confirmed, showing unimodal and symmetrical peaks with low poly-dispersity (Fig. 3i and Supplementary Fig. 26). Furthermore, unlike previous literature report[12], this photopolymerization mediated by GOx containing the cofactor FAD, does not involve the catalytic process related to the cofactor that need to consume the substrate, and it was mediated by Tyr residues in protein without the presence of substrate. The above results showing the versatility of the employed proteins, should indicate it is very likely that other Tyr residues containing proteins might also be available for initiating photopolymerization.

The distinctive advantages of such developed protein-initiated photopolymerization lies in the easy synthesis of protein-polymer bio-conjugates with good biocompatibility. Currently, there is a growing interest in the widely reported protein-polymer bioconjugate, due to its promising alternative for enhancement of enzyme efficacy and ther-apeutic delivery. While given the low modification efficiency for graft-to approach, as well as the removal of initiator in graft-from approach having adverse influence on protein activity, such developed poly-merization method, which could utilize macromolecule chain transfer agent BSA-CTA as both the initiator and chain transfer agent for in-situ polymerization to synthesize BSA-polymer bioconjugates, can be proved to be an efficient way. The evidence for BSA-CTA synthesis were summarized in Supplementary Fig. 27, and by quantifying the UV-Vis absorption spectra of BSA-CTA, the amount of CTA modification was about 3.1 CTA per BSA (Supplementary Table 9). Then, the BSA-poly(**2**) bioconjugates were synthesized after photopolymerization as shown in Supplementary Fig. 28a, b, and the SDS-PAGE showed that the poly-merization process resulted in the decreased concentration of BSA-CTA and the formation of new substance with higher molecular weight, which verified the successful initiation of the polymerization by BSA-CTA, with the generation of BSA-poly(**2**) that could then be successfully used as building blocks to prepare proteinosomes with the diameter in the range of 20–40 μm (Supplementary Fig. 28c, d)[62,63].

## Yeast cells surface or intracellular polymerization

Significantly, in comparison with the conventional organic photo initiator, the biocompatibility of this studied polymerization system was greatly improved. To demonstrate so, we employed yeast cells (*S. cerevisiae*) as a model. By performing the BSA-mediated RAFT photo-polymerization in the cell mixture solution containing BSA, **6** and **3** under 405 nm light, the viability of the yeast cells was over 95%, accompanied with the generation of poly(**3**) with the unimodal GPC curves (Supplementary Figs. 29, 30). Accordingly, by anchoring CTA onto the surface of yeast cell, the in-situ graft of different polymers from the yeast cell surface was realized for cellular phenotype manipulation (Fig. 4a). Using **5** as the monomer, the polymerization modification was processed for 1.5 h under 405 nm LED light. The

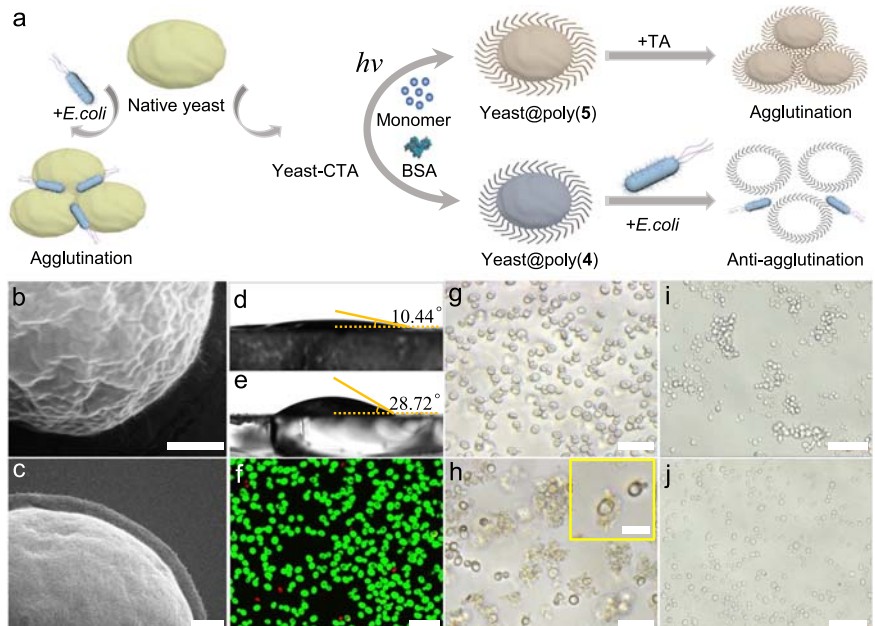

**Fig. 4 | BSA-mediated RAFT photopolymerization on the surface of yeast cells for agglutination/anti-agglutination behavior manipulation. a** Schematic illustration of yeast cells surface modified by in-situ polymerization of **4** or **5** for cellular phenotype manipulation. SEM images of yeast cells (**b**) before and (**c**) after poly(**5**) modification. Scale bars in (**b, c**), 500 nm. Contact angle images of (**d**) native yeast cells and (**e**) yeast@poly(**5**) cells. **f** Confocal fluorescent microscopy image of yeast@poly(**5**) cells stained by FDA (green) and PI (red). Optical microscopy images of BSA-mediated yeast-CTA cells surface-initiated polymerization of **5** with tannic acid (TA) treatment, (**g**) 0 h and (**h**) after polymerization for 1.5 h, and yeast cells agglutination can be observed due to the strong affinity of poly(**5**) with tannic acid (TA). The yeast cells were collected by centrifugation and washed three times with deionized water. The inset in (**h**) is the zoomed optical microscopy image. Scale bars in (**f–h**), 25 μm. Scale bars in the inset in (**h**), 10μm. **i, j** *E. coli* induced agglutination/anti-agglutination assay, (**i**) native yeast cells and (**j**) yeast@poly(**4**) cells. Scale bars in (**i, j**), 50 μm. The experiments were independently repeated three times with similar results in (**f–j**).

successful modification of poly(**5**) onto the yeast cell surface was clearly observed from the SEM image compared with the native yeast cells (Fig. 4b, c). Meanwhile, after poly(**5**) modification, the hydrophilicity of the yeast cells was varied, as indicated by the altered contact angle from 10.44° to 28.72° (Fig. 4d, e). The result of FDA assay was shown in confocal fluorescent microscopy image of yeast@poly(**5**) stained by fluorescein diacetate (FDA, stained living cells, green) and propidium iodide (PI, stained dead cells, red), illustrating the viability of yeast cells can retain as high as 96% (Fig. 4f). We added tannic acid (TA) into yeast@poly(**5**) cells aqueous suspension, which was obtained by centrifuging BSA/yeast-CTA/**5** solution after polymerization for 1.5 h and washed three times with deionized water. Notably, due to the strong affinity between the poly(**5**) on yeast cell surface and TA via hydrogen bond interactions[64], the apparent agglutination of yeast cells was observed after polymerization, while no similar behavior was present before polymerization (Fig. 4g, h). Moreover, it is well known that yeast cells could aggregate rapidly when they were mixed with *E. coli* due to the recognition of yeast cell wall by α-D-mannose-binding proteins on the surface of *E. coli* cilia[65]. Grafting polymer poly(**4**) from yeast cells wall was also achieved, yet the obtained yeast@poly(**4**) cells exhibited enhanced anti-agglutination performance against *E. coli* compared with native yeast cells, due to the dense polymer layer on the cells surface blocking the binding between yeast cells and *E. coli* (Fig. 4i, j). Overall, the obtained results should indicate that such developed photopolymerization system was applicable for functionalizing yeast cells with high cellular viability.

To be of great interest, it was also observed that in the presence of high concentration of proteins, such as over 0.9 mM BSA, the photopolymerization of **1** could be achieved at normal aerobic condition with induction periods of the polymerization at 1.5–2 h (Supplementary Fig. 31). Accordingly, such photopolymerization could be well proceeded at aerobic condition in various protein-containing biological media including yeast extract (YE) and yeast

extract peptone dextrose medium (YPD) (Supplementary Figs. 32, 33). Successful homogeneous and heterogeneous polymerization mediated by biological medium in the presence of oxygen were implemented to synthesize a range of controlled molecular weight polymers, which expanded the application prospect of this method in the field of bioengineering material synthesis. Specially, given the diversity of proteins in living organisms, such developed photopolymerization system would then provide a new strategy to realize the non-natural polymer synthesis inside the biotic environment.

As a proof-of-concept demonstration, we further explored the polymerization inside yeast cells (average amounts of protein, 29–65% on a dry weight). Firstly, the cell lysates obtained by lyophilization and dispersed into H₂O were mixed with **6** and **3**. As expected, successful RAFT photopolymerization was performed to reveal the possibility of yeast cells-mediated polymerization (Supplementary Fig. 34). Then, 4-vinylbenzenesulfonic acid sodium salt (Nass) and fluorescein O-methacrylate (FMA) with green fluorescence were employed as the model monomers. Yeast cells were co-cultured with Nass (98 mM) and FMA (0.5 mM) for 6 h and then washed three times with deionized water. Yeast cells loaded with Nass and FMA were treated under 405 nm LED light for 40 min and untreated with light as a control. In view of the fact that the green fluorescence of yeast cells faded away along the cell metabolism, while intracellular polymerization would slow down the process due to the formation of high molecular weight polymer inside cells. Thereby, yeast cells loaded with monomers after being treated or untreated with light for 40 min, were kept on culturing for 134 h and observed at 0, 96 and 134 h, respectively (Fig. 5a). Notable stronger fluorescence of cells after 405 nm light treatment was exhibited in comparison with that of control yeast cells at 96 h. The fluorescence intensity statistics of yeast cells also showed significant difference ($p < = 0.001$) in the experimental group and control group at 96 h (Fig. 5b), indicating the successful polymerization inside the yeast cells

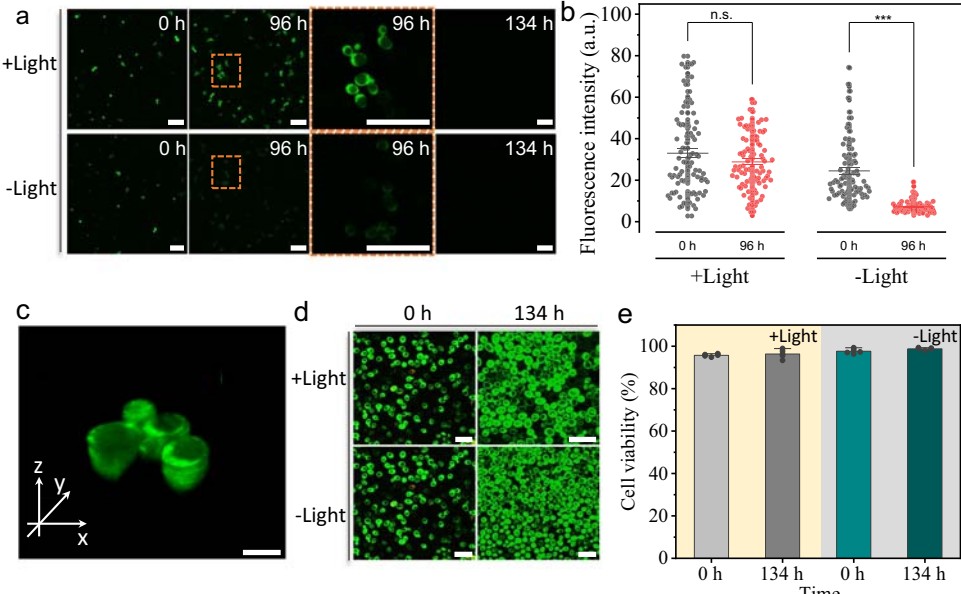

**Fig. 5 | The photopolymerization inside living yeast cells. a** The confocal fluorescence microscopy images of yeast cells. The yeast cells loaded with Nass and FMA (green) after treated or untreated with 405 nm light for 40 min, were continued to culture for 134 h and observed at 0 h, 96 h and 134 h, respectively. Scale bars 25 μm. The experiments were independently repeated three times with similar results. **b** The corresponding fluorescence intensity statistics of the yeast cells at 0 h and 96 h. Data are presented as mean ± s.e., $n = 92$, n.s. not significant; ***$p < = 0.001$ (Two-Way ANOVA). **c** 3D confocal fluorescence image of yeast cells loaded with Nass and FMA (green) at 96 h after treated with 405 nm light, stained by PI (red). Scale bar, 5 μm. **d** Confocal fluorescent microscopy images of yeast cells loaded with Nass and FMA (green) at 0 h and 134 h after treated or untreated with 405 nm light, stained by FDA (green) and PI (red). Scale bars, 16 μm. The experiments were independently repeated three times with similar results. **e** The corresponding cell viability statistics of yeast cells at 0 h and 134 h in (**d**). Data are presented as mean ± s.d. ($n = 5$).

accompanied with the complete metabolism of fluorescent substances at 134 h. What is more important, by staining the yeast cells with FDA and PI dye, the maintenance of cells viability after loading monomers was well observed from the 3D confocal fluorescence image of yeast cells at 96 h (Fig. 5c, Supplementary Fig. 35), and both yeast cells treated or untreated with 405 nm LED light showed highly green fluorescent from FDA, indicating that the yeast cells still maintained normal proliferation, and more than 95% cell viability was observed after 134 h (Fig. 5d, e, Supplementary Fig. 36).

In conclusion, our results describe a feasible strategy to carry out photopolymerization in biologically related aqueous solution by specially using the Tyr residues containing protein as a photo initiator. This highly universal approach has allowed for successful generation of a series of polymers with well controllable molecular weight and narrow polydispersity, by employing a variety of proteins (BSA, HSA, Lysozyme, RNase-A and GOx) and monomers (**1, 2, 3, 4, 5**). We further revealed the proton-coupled electron transfer mechanism in the photopolymerization process under the light of 405 nm, the generated excited-state TyrOH* residue in protein transferred electron to CTA or monomer along with the formation of oxidized-state Tyr-ox. Significantly, the developed system also shows good biocompatibility, and the non-natural polymers could be successfully generated both on the surface of yeast cell with improved hydrophobicity or agglutination/anti-agglutination manipulation and inside the yeast cell. The simple operation, good biocompatibility, as well as the controllability in the molecular weight distribution of this approach are key features that make it a powerful option for the efficient functional modulation of living organism in vitro and in vivo by generating various non-natural polymers. We foresee future applications of the Tyr containing proteins initiated photopolymerization will bridge cofactor-free protein with the controlled radical polymerizations (CRP), as well as addressing important challenges in intra-organism polymerization and bioinspired living material synthesis.

## Methods

### General procedure for the kinetic study of monomer 3 in H₂O

Typically, a glass vial (5 mL) charged with BSA (62.76 mg, 0.9443 μmol), **3** (300 mg, 3.0263 mmol), **6** (15.4 mg, 6.5709 μmol), and deionized water (3 mL) at a molar ratio of [BSA]:[**6**]:[**3**] = 0.144:1:460 was sealed with a rubber septa. The reaction mixture was degassed by bubbling argon for 25 min. General procedure of degassing oxygen: prepare a long needle and a short needle to be inserted into the vial sealed with a rubber septa. The long needle was fully inserted into the solution and the short needle was placed in the air section at top of the vial. The other end of the long needle was connected with the argon cylinder to inject argon into the vial, and the other end of the short needle was connected with the air to release oxygen from the vial. The solution of reaction mixture was degassed by bubbling argon for 25 min to completely drain the oxygen in the vial. Then, the vial was irradiated under violet LED light (20 W, 405 nm) at room temperature. Aliquots were taken at predetermined time intervals via degassed syringe. Each reaction aliquot was dried by freeze-drying and was dissolved in DMF overnight to ensure that the polymer was completely dissolved, while the protein precipitation was removed by 0.45 μm filter for GPC characterization to measure number-average molecular weight ($Mn$) and the polydispersity ($Mw/Mn$). Monomer conversion was determined by ¹H NMR spectrum.

### General procedure for the ON/ OFF study of monomer 3 in H₂O

The process of temporal control was similar to the method described in the kinetic study section. Specially, the LED light on-off was switched at predetermined time points. The vial was covered with foil and placed in the dark condition when the LED light was turned off. Aliquots were taken at predetermined time intervals via degassed syringe. Monomer conversion was determined by ¹H NMR spectrum. Each reaction aliquot was dried by freeze-drying and dissolved in DMF for GPC characterization to measure number-average molecular weight ($Mn$) and polydispersity ($Mw/Mn$).

## General procedure of photopolymerization in yeast cells

Typically, employing Nass and FMA as the model monomers, yeast cells were obtained by culturing in YPD medium for 24 h at 30 °C. Yeast cells were firstly washed three times with deionized water by centrifugation and resuspended with YPD medium. Then, the yeast cells ($OD_{600}$ = 30) were co-cultured with Nass (98 mM) and FMA (0.5 mM) in YPD medium under shaking at 130 rpm for 6 h at 30 °C, and then washed three times with deionized water and resuspended in $H_2O$ to obtain yeast cells loaded with Nass and FMA. The yeast cells loaded with Nass and FMA in $H_2O$ ($OD_{600}$ = 30, 1 mL) were degassed by bubbling argon for 25 min, and was illuminated under violet LED light (20 W, 405 nm) at room temperature for 40 min and untreated with light as a control. The yeast cells were observed under confocal fluorescent microscopy, and using ImageJ for fluorescence intensity statistics of yeast cells.

## Reporting summary

Further information on research design is available in the Nature Portfolio Reporting Summary linked to this article.

## Data availability

The experimental data generated in this study are available as the Supplementary Information. Source data are provided with this paper. All other relevant data are available from the corresponding authors upon request. Source data are provided with this paper.

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

## Acknowledgements

We thank the NSFC (22171058 (X.H.) and 51873050 (X.L.) and the Fundamental Research Funds for the Central Universities (HIT.O-CEF.2021027 (X.H.)) for financial support.

## Author contributions

M.Z. and X.H. conceived the experiments; M.Z., S.W., Z.L. performed the experiments; M.Z., S.W., Z.L., J.L., Z.X., X.L., and X.H undertook data analysis; M.Z., X.L., and X.H. wrote the manuscript.

## Competing interests

The authors declare no competing interests.
