## [Peer Review File · Nature Communications]

Tyrosine residues initiated photopolymerization in living organismsReviewers' Comments:

Reviewer #1:

Remarks to the Author:

This communication reports a photo activation of RAFT process using tyrosine containing proteins, such as BSA or GOx. It was discovered that the excited-state oxidation potential of Tyr containing proteins is estimated to be -3.38 V, which claimed as sufficient to carry out a PET-RAFT-like activation. Radical mechanism was investigated with trapping experiments. Other 3 proteins were also confirmed as scaled PCET catalyst. The paper was then described attaching RAFT group on to the cell surfaces, and the grating from RAFT polymerization was occurred on the cell surface. The manuscript further described how polymerization inside of cell may occur.

There are certainly novelties in this manuscript worth considering publication for Nature Communication; for example, taking natural proteins as catalyst in RAFT process, although this is known, but it is with more in-depth investigations. Attaching RAFT agent on to proteins or cell is certainly another novel approach. However, there are deficient sections the authors should address before the acceptance of the manuscript.

1. How easy is the visible light activated process as described? Is the intensity of light very relevant? Is there oxygen degassing issues? My understanding is that such process is not easy to proceed and can the authors put more detailed on this; so the published work become repeatable by other labs.
2. If radical mechanism is confirmed, the next obvious question is that radical can activate RAFT process by radical speciate just like normal radicals do. On the other hand, the reducing power of radical species or equivalent may also dissociate RAFT agent to proceed the PECT type polymerization. Which process is more likely and is the evidence very clear?
3. The evidence of polymerization inside of cell is not strong or sufficient. It was not clear if RAFT agent was used judge from the description. Secondly, as monomer is fluorescent tagged, it is hard to tell from the image if the color from the image is due to the monomers or polymers. Can other experiment such as quenching of fluorescent color be used to confirm the polymerization inside of the cell beyond doubt? Or can the polymer be isolated from the experiments?

Reviewer #2:

Remarks to the Author:

In this manuscript, Huang and co-workers describe the use of tyrosine-containing proteins to mediate RAFT polymerization under violet LED light. Polymerizations of several monomers were conducted in solution, from the surface and inside of yeasts. The proton coupled electron transfer (PCET) mechanism between the excited-state TyrOH* residue in proteins and the monomers or the chain transfer agents were proposed. This is an interesting work considering that tyrosine-protein photopolymerization is a new addition to the toolbox of controlled radical polymerization. However, the disadvantage of this polymerization system is the use of a large amount of protein (~16% mass percent relative to monomer) and the difficult of polymer separation from protein. Another weakness is that actually the protein structure is not necessary for the polymerization (as denatured proteins also worked) and the tyrosine moiety is responsible for the polymerization. It is not obvious what the advantages of using proteins instead of tyrosine. In addition, the proteins function only as initiators, not as catalysts (at least the authors did not convincingly prove they are catalysts), because there is no pathway to restore the original tyrosine from the proposed mechanism. Overall, the manuscript provides an interesting new method for photopolymerization and I would encourage a major revision for the authors to address the many detailed points as listed below.

Why a 405 nm LED light was chosen? It seems BSA almost does not absorb at 405 nm.

It seems this polymerization suffers relatively low efficiency because a high concentration of protein is

needed (0.315 mM BSA vs 10% DMA). A rough calculation indicates that the mass ratio of BSA/monomer is $\sim 16\%$ (based on the data in Figure S1 and taking the M_n of BSA 66 kg/mol), which suggests a fair large amount of protein is needed to enable a higher monomer conversion (83%) and the resulting polymer is contaminated with a large amount of protein. This is also problematic for GPC measurements of the resulting polymers, which means efforts should be taken to effectively remove BSA prior to GPC measurements. However, it seemed no such separation effort was taken after the polymerization.

Page 3, it was not clear which polymerization system was used for on/off experiments. Such information should be provided in the text and in the Figure captions. Similarly, the polymerization information for each of the panels shown in Figure 2 should be clearly indication in the figure caption.

Figure 2d, if a line for theoretical molecular weight is also shown, it would be helpful to see the degree of matching between theoretical molecular weights and GPC measurements.

Figure 2e, the particles seemed to be flattened on the substrate surface in the SEM picture. This is probably because poly(MEA) has a low glass transition temperature (lower than room temperature, which is probably the temperature at which SEM was taken). Thus, the particle size and morphology measured by SEM does not reflect the actual size and morphology of the particles. What data was used for particle size evolution against time, DLS or SEM? This information should be clearly indicated.

Figure 2h, the plot of M_n vs time is not very meaningful as it simply reflects the results of Figure 2g. A more meaningful plot is $\ln(M_0/M_t)$ vs time, which shows whether first order can still be obtained or not during on/off process.

In the synthesis of PDMA1-b-PDMA2, the information on the synthesis of PDMA1 (PDMA500) should be provided: what was the monomer conversion and how was the polymer purified? Also, such information should be provided for the chain extension step: what was the monomer conversion of the second step of polymerization.

The fluorescence of BSA was quenched by CTA and monomers at different efficiency. Could the trend of the different quenching ability of monomers be reflected by polymerization rates? That is, could higher polymerization rate be obtained for monomers with a higher quenching ability, especially for monomers with similar polymerization rate coefficient?

The thiol product of CTA in Figure 3e need to be demonstrated in detail, as well as how the CTA maintains the controlled polymer synthesis despite the decomposition into thiols. It seems there is no pathway to regenerate the original tyrosine structure.

In the PTIO experiments, it is helpful to present the UV-vis spectrum of PTIO and its reaction in the presence of BSA. The authors described this type of reaction as an electron transfer process. As I understand, PTIO is a radical trap, which captures a generated radical to form an adduct. Then, the question is: what is identity of the radical generated? EPR experiments would be helpful to identify the generated radical. To clarify this point, either suitable references should be provided on electron transfer to PTIO or EPR should be taken.

The authors showed that the 3D structure of protein was not necessary for the polymerization because aggregated and denatured BSA could also initiate the polymerization. Then, the question is why protein was used for the polymerization (a lot of mass is wasted), not simply tyramine (the amine group needs to be protected in this case to avoid aminolysis of RAFT agent)?

Information of the amounts of other proteins or their molar ratios relative to CTA and monomers used in polymerization should be given.

Figure S22f, what were the molecular weights and dispersity of the polymers.

It is not surprising that photopolymerization proceeded under aerobic conditions when high concentrations of proteins were used, since some of the generated radicals consume oxygen. However, it is important to evaluate how the polymerization was affected by the presence of air. Are there any induction periods observed due to initial consumption of oxygen? How are the quality of the obtained polymers affected by the presence of oxygen and the excess use of protein?

I donot understand why in the absence of light highly fluorescent was still observed in Figure 5d.

Photos of photopolymerization experimental setup can be provided in the supporting information.

Reviewer #3:

Remarks to the Author:

The presented paper deals with photo-induced polymerization using different Tyr-containing proteins as polymerization initiators (called by authors as a potential new biocompatible polymerization system). This is an interesting idea that could been used further in research works. The work seems to be complete and authors applied plethora of analytical methods including kinetic analysis and cell imaging. Experiments appear to be carried out professionally, but I have a filling, that the structure of presented results is messy and overloaded with collected data, thus it is difficult to follow. However, in my opinion, I have doubts, if the data interpretation and proposed explanation are correct. In my knowledge, and also as presented in UV-Vis spectra in Supplementary data, tyrosine and proteins containing Tyr would not absorb light, especially to form the excited singlet state, at 405 nm. In fact, I would feel more convinced, if authors can present the UV vis spectra of the mixture protein/monomers applied in experiments together with separated UV-Vis for monomers and proteins to check, if there is any interactions before (ground state complex?). Other vise, authors should think again about the possible explanation of polymerization occurring in this system.

Response to the reviewers' comments (Reviewer#1)

Comments to the Author:

This communication reports a photo activation of RAFT process using tyrosine containing proteins, such as BSA or GOx. It was discovered that the excited-state oxidation potential of Tyr containing proteins is estimated to be -3.38 V, which claimed as sufficient to carry out a PET-RAFT-like activation. Radical mechanism was investigated with trapping experiments. Other 3 proteins were also confirmed as scaled PCET catalyst. The paper was then described attaching RAFT group on to the cell surfaces, and the grating from RAFT polymerization was occurred on the cell surface. The manuscript further described how polymerization inside of cell may occur.

There are certainly novelties in this manuscript worth considering publication for Nature Communication; for example, taking natural proteins as catalyst in RAFT process, although this is known, but it is with more in-depth investigations. Attaching RAFT agent on to proteins or cell is certainly another novel approach. However, there are deficient sections the authors should address before the acceptance of the manuscript.

Response: We thank Reviewer#1 for the assessment of our manuscript, valuable suggestions as well as the careful checking. The manuscript has been carefully revised according to your professional suggestions. All of the modifications have been highlighted with yellow colour in the revised manuscript and Supplementary Information. All of your professional comments are seriously considered and replied in detail as follows.

Comment 1: How easy is the visible light activated process as described? Is the intensity of light very relevant? Is there oxygen degassing issues? My understanding is that such process is not easy to proceed and can the authors put more detailed on this; so the published work become repeatable by other labs.

Response: Many thanks for this instructive and professional comment. We have carefully considered and responded including the effect of light intensity on polymerization and the oxygen degassing issues as follows.

(1) The effect of light intensity on polymerization.

In general, the polymerization process is easily activated by visible light which could be triggered within half an hour with 7.2% conversion at the light intensity of 1.38 mW/cm² (Fig. 2b). We performed the additional experiments, and the relevance between the RAFT polymerization and light intensity was added as new Supplementary Fig. 11 and Supplementary Table 5. The RAFT polymerization rate increases with the increase of light intensity. The monomer conversion could increase from 53.7% to 91.8% when the light intensity increases from 0.8 to 4.3 mW/cm².

(2) The oxygen degassing issues.

The experimental procedure to degass oxygen has been added into Methods section which is now reading: "Prepare a long needle and a short needle to be inserted into the vial sealed with a rubber septa. The long needle was fully inserted into the solution and the short needle was placed in the air section at top of the vial. The other end of the long needle was connected with the argon cylinder to inject argon into the vial, and the other end of the short needle was connected with the air to release oxygen from the

vial. The solution of reaction mixture was degassed by bubbling argon for 25 minutes to completely drain the oxygen in the vial.”

Modification: According to the reviewer’s suggestions, we have added the experiments about the effect of light intensity on polymerization and the detailed description of the oxygen degassing into revised manuscript in new Supplementary Fig. 11 and Supplementary Table 5 and Method section.

New Supplementary Fig. 11. The GPC traces of PDMA synthesized at different light intensity. [DMA] = 10% w/v, [BSA] = 0.315 mM, [BSA]:[CTA]:[DMA] = 0.125:1:400, 6 h, room temperature, Ar protection.

New Supplementary Table 5. RAFT photopolymerization of DMA at different light intensity.^[a]

entry	Intensity (mW/cm ²)	M_n (kg/mol)	M_w/M_n	Conversion
1	0.8	102.6	1.1	53.7
2	1	102.4	1.1	63.04
3	1.9	101.4	1.11	79.38
4	3.6	102	1.12	85.99
5	4.3	100.4	1.11	91.8

^[a] Polymerization conditions: [DMA] = 10% w/v, [BSA] = 0.315 mM, [BSA]:[CTA]:[DMA] = 0.125:1:400, 6 h, room temperature. Conversion ratio was determined by ¹H NMR spectrum in D₂O.

Comment 2: If radical mechanism is confirmed, the next obvious question is that radical can activate RAFT process by radical specie just like normal radicals do. On the other hand, the reducing power of radical species or equivalent may also dissociate RAFT agent to proceed the PECT type polymerization. Which process is more likely and is the evidence very clear?

Response: Many thanks for the thoughtful comment. We have fully considered this comment and replied to it by performing five experiments to further confirm the tyrosine-based proton-coupled electron transfer (PCET) process beside the original fluorescence quenching experiments based on Stern-Volmer plotting to confirm the electron transfer between BSA and PEG-CTA or monomers.

- (1) This work proposed the electron transfer (ET) between protein and monomer or chain transfer agent (CTA) coupled to proton transfer (PT) by tyrosine-based proton-coupled electron transfer (PCET) reactions. The photopolymerization involved two processes: (i) generation of initial radical by PCET; (ii) process of chain propagation including triggering the general radical polymerization or RAFT process. Upon excitation with violet light, excited-state TyrOH* residue in protein transferred the electron to monomer, producing initial radical to trigger chain propagation. When the chain transfer agent was added, excited-state TyrOH* residue in protein would rapidly transfer the electron to CTA or monomer, generating initial radical to activate RAFT process. The image in Fig. 3e was modified and the whole polymerization procedure was shown.
- (2) Confirm the tyrosine-based PCET mechanism:
- (i) The fluorescence quenching experiment of Ribonuclease A (RNase-A) without containing tryptophan residues was investigated referred to new Supplementary Fig. 20. The maximum emission wavelength of RNase-A is 312 nm ($\lambda_{ex} = 280$ nm) mainly contributed by Tyr. With the addition of PEG-CTA or monomer DMA, respectively, the gradual decrease of fluorescence intensity and red shifts was clearly observed, which indicated excited state ionization of Tyr and formation of TyrO[•] radical.
- (ii) The photocurrent responses experiment of BSA and Tyr were carried out along with the 405 nm light switch. BSA and Tyr were immobilized by dropping 100 μ L of 60 mg/mL BSA and 3.27 mg/mL Tyr suspension in 0.5% Nafion solution onto a FTO substrate surface, respectively, corresponding to 20 Tyr on each BSA. BSA and Tyr showed the similar photocurrent responses against light ON and OFF, which indicated the BSA and Tyr have the similar ability of electron transfer under 405 nm light (new Supplementary Fig. 18a).
- (iii) Blocking experiments of the tyrosine residues in BSA were performed. According to previous reports (*Arch. Biochem. Biophys* 1999, **362**, 105-112), the tyrosine residues both “free” and “buried” tyrosyl residues in BSA were blocked by iodine resulting in the loss of the EPR signals from tyrosine-derived radicals (Equation 1). The “free” tyrosine residues on BSA were blocked using N-Acetylimidazole (new Supplementary Fig. 18b), and then acetylated BSA, iodinated BSA and native BSA (using pH 9.5 buffer to treat the same process) were utilized for photopolymerization of MEA in H₂O, respectively. Compared with native BSA, the monomer conversion of polymerization mediated by acetylated BSA and iodinated BSA decreased as shown by the turbidity study in new Supplementary Fig. 18c.
- $$\text{Tyrosine} + 2 \text{I}_2 \longrightarrow 3, 5\text{-diiodotyrosine} + 2\text{I}^- + 2\text{H}^+ \quad (\text{Equation 1})$$
- (iv) After polymerization, the amounts of Tyr residues on BSA were determined by the method reported by J. CHRASTIL (*Anal. Biochem* 1986, **158**, 443-446). The number of tyrosine was determined by UV-vis absorption at 355 nm. As shown in new Supplementary Fig. 18d, it revealed a decrease in the Tyr number on BSA due to oxidation after 9 hours of polymerization. Base on the above analyses and comparisons including excited state ionization of Tyr, the similar photocurrent responses of BSA, the lower rate of polymerization mediated by BSA blocking tyrosine residues and the decrease amount of Tyr residues on BSA after polymerization, we inferred that protein

transferred the electron to monomer or CTA by tyrosine-based PCET reactions. Moreover, it should be mentioned that since there are 20 Tyr on each BSA, BSA can be reused for the cyclic polymerization, which was exhibited in new Supplementary Fig. 19.

Modification: According to the reviewer's comments, we have revised manuscript in main text page 5 paragraph 1 and in Method section with yellow highlighted parts which read as follows, as well as the attached new Supplementary Fig. 18, 19, 20.

"...It is recognized that Tyr is one of the main reaction sites during the electrolytic oxidation or photooxidation process of BSA⁵²⁻⁵⁴, and the similar photocurrent responses of BSA and Tyr were shown under 405 nm light (Supplementary Fig. 18a)."

"...This was further confirmed by a control experiment that Tyr residues in BSA were blocked by forming acetylated BSA and iodinated BSA, respectively, which was then used to mediated photopolymerization of MEA, respectively, showing lower conversion compared with native BSA (Supplementary Fig. 18b, c)⁵⁶. While due to the oxidation of Tyr, a decrease in the Tyr number on BSA was observed after 9 hours polymerization (Supplementary Fig. 18d). However, given that there are 20 Tyr on each BSA, BSA can still be reused for three times cyclic polymerization (Supplementary Fig. 19). Moreover, the direct interaction between the fluorophore Tyr residues in other protein and PEG-CTA/DMA was investigated by fluorescence quenching experiment of Ribonuclease A (RNase-A) without containing tryptophan residues (Supplementary Fig. 20). The maximum emission wavelength of RNase-A is 312 nm ($\lambda_{ex} = 280$ nm) mainly contributed by Tyr. Along with the increasing of the PEG-CTA/DMA concentration, the obvious decrease of the emission intensity and red shifts were clearly observed, which then resulted in excited state ionization of Tyr and TyrO \cdot radical formation."

"...FTO as working electrode was used in experiment of photocurrent responses in 0.3 M Na₂SO₄ solution."

New Supplementary Fig. 20. Fluorescence quenching emission spectrum of Ribonuclease A with the quencher of different concentrations, (a, b) DMA and (c, d) PEG-CTA. [RNase-A] = 0.128 mM.

New Supplementary Fig. 18. The study of the Tyr residues on BSA mediated polymerization. (a) The Photocurrent responses of BSA and Tyr with the 405 nm light switch. BSA and Tyr were immobilized by dropping 100 μL of 60 mg/mL BSA and 3.27 mg/mL Tyr suspension in 0.5% Nafion solution onto a FTO substrate surface, respectively, and evaporating the solvent at room temperature. Blocking experiments of the Tyr residues in BSA including acetylated BSA, iodinated BSA and native BSA (using pH 9.5 buffer to treat the same process) were used to mediate the photopolymerization of MEA^{3,4}. (b) The UV-Vis absorption spectra of acetylated BSA, iodinated BSA and native BSA (treated pH 9.5 buffer). (c) The turbidity study of polymerization of MEA mediated by acetylated BSA, iodinated BSA and control BSA (treated pH 9.5 buffer) for 6 h, characterized by the visible light absorption intensity at 650 nm. According to previous reports, the Tyr residues both “free” and “buried” tyrosyl residues in BSA were blocked by iodine. The “free” Tyr residues on BSA were blocked using N-Acetylimidazole. Data are presented as mean \pm s.d., error bars indicate standard deviations ($n = 4$). (d) The determination of the amount of Tyr residues on BSA after polymerization for 9 h. The number of Tyr residues on BSA were determined by the method reported by J. CHRASTIL⁵. The number of Tyr was determined by UV-vis absorption at 355 nm. Data are presented as mean \pm s.d., error bars indicate standard deviations ($n = 3$). [BSA] = 0.315 mM, [M] = 10% w/v, violet LED light (20 W, 405 nm), room temperature, Ar protection.

New Supplementary Fig. 19. Plot of conversion ratio of photopolymerization of MEA mediated by recycled BSA. The inset was schematic illustration of BSA-mediated cycled photopolymerization of MEA.

Comment 3: The evidence of polymerization inside of cell is not strong or sufficient. It was not clear if RAFT agent was used judge from the description. Secondly, as monomer is fluorescent tagged, it is hard to tell from the image if the color from the image is due to the monomers or polymers. Can other experiment such as quenching of fluorescent color be used to confirm the polymerization inside of the cell beyond doubt? Or can the polymer be isolated from the experiments?

Response: Many thanks for the professional and thoughtful comment. We have carefully considered and responded as follows. I am sorry for the confusion that no RAFT reagents were added into the photopolymerization system inside living yeast cells in Fig. 5. By using a large amount of monomers (as high as 200 mM) to perform the polymerization, then we tried to isolate the polymers from the cells, while only a weak GPC peak was observed. Therefore, we chose to utilize the metabolic differences between polymer (slower metabolism) and small molecules (faster metabolism) in yeast cells. Yeast cells firstly were loaded with Nass and FMA (green fluorescent monomer), which were then treated with 405 nm LED light for 40 min and untreated with light as a control. Along with illumination, the fluorescent monomers (FMA) copolymerized with monomer Nass in yeast cells, forming high molecular weight polymers, resulting in slower metabolism. Therefore, when yeast cells with polymerization (treated light) and control cells (untreated light) were continued to culture for a long time, the former still exhibited fluorescence due to the slower metabolism of fluorescent copolymers, while the latter showed almost no fluorescence due to the early expulsion of small molecule fluorescent monomers (Fig. 5a, b). In addition, according to the reviewer's suggestion, to further confirm so, cell lysates including cell lysates of yeast cells and NIH-3T3 cells (mouse-sourced stem cells) were used to perform RAFT photopolymerization. As shown in **new Supplementary Fig. 33**, the cell lysates were mixed with PEG-CTA and monomers to mediate the photopolymerization of DMA or MEA, respectively, where successful polymerization was achieved to form PDMA or PME A with narrow molecular weight distribution.

Modification: According to the reviewer's comments, we have clarified the polymerization procedures in the main text and the Method section. The experiments of RAFT photopolymerization of monomers mediated by cell lysates of yeast cells and NIH-3T3 cells were added as **new Supplementary Fig. 33**, and the detailed descriptions of culture of NIH-3T3 cell have also been provided in the Method section in Supplementary Information, which read as follows:

New Supplementary Fig. 33. Cell lysates mediated photopolymerization. (a) Microscopy image of native yeast cells. The inset was photograph of yeast cells lysates. (b) Microscopy image of native NIH-3T3 cells (mouse-sourced stem cells). The inset was photograph of NIH-3T3 cells lysates. (c) GPC traces of RAFT photopolymerization of DMA, mediated by yeast cells lysates. (d) GPC traces of RAFT photopolymerization of MEA, mediated by NIH-3T3 cells lysates. The inset was photograph of photopolymerization of MEA for 10 h. Cells lysates were obtained by lyophilization and dispersion into H₂O. The synthesized PDMA and PMEA with narrow molecular weight distribution were observed. [Yeast cells lysates] = 5 mg/mL. NIH-3T3 cells cultured to the cell density shown in (b) in T25 cell culture flask were collected, and then lyophilization and dispersion in 1 mL of H₂O, [PEG-CTA] = 1.7 mM, [MEA] = 10% w/v, [DMA] = 10% w/v, violet LED light (20 W, 405 nm), room temperature, Ar protection.

Cell culture of 3T3 cell

NIH-3T3 cells were cultured in Dulbecco's Modified Eagle Medium (DMEM), containing with 10% (vol/vol) fetal bovine serum and 1% (vol/vol) penicillin/streptomycin at 37°C and 5% CO₂, and then regularly replaced with the fresh medium.

Response to the reviewers' comments (Reviewer#2)

Comments to the Author:

In this manuscript, Huang and co-workers describe the use of tyrosine-containing proteins to mediate RAFT polymerization under violet LED light. Polymerizations of several monomers were conducted in solution, from the surface and inside of yeasts. The proton coupled electron transfer (PCET) mechanism between the excited-state TyrOH* residue in proteins and the monomers or the chain transfer agents were proposed. This is an interesting work considering that tyrosine-protein photopolymerization is a new addition to the toolbox of controlled radical polymerization. However, the disadvantage of this polymerization system is the use of a large amount of protein (~16% mass percent relative to monomer) and the difficult of polymer separation from protein. Another weakness is that actually the protein structure is not necessary for the polymerization (as denatured proteins also worked) and the tyrosine moiety is responsible for the polymerization. It is not obvious what the advantages of using proteins instead of tyrosine. In addition, the proteins function only as initiators, not as catalysts (at least the authors did not convincingly prove they are catalysts), because there is no pathway to restore the original tyrosine

from the proposed mechanism. Overall, the manuscript provides an interesting new method for photopolymerization and I would encourage a major revision for the authors to address the many detailed points as listed below.

Response: We sincerely thank the reviewer's positive comments on our work. Your professional suggestion is helpful for the improvement of this article. The manuscript has been carefully revised according to your suggestions. All of the modifications have been highlighted with yellow colour in the revised manuscript and supplementary information. We have carefully replied to your comments point-to-point as follows, which we hope meet with your approval again.

Comment 1: Why a 405 nm LED light was chosen? It seems BSA almost does not absorb at 405 nm.

Response: In this study, we aim to search for a mild and biocompatible polymerization system which should be a key point to realize intrinsically non-natural macromolecules synthesis for modulating living organism function/behaviour. Accordingly, to keep the biocompatibility of the polymerization system at most, a 405 nm LED light with a weak light intensity (1.38 mW/cm^2) is chosen finally with compromised polymerization rate despite proteins have the strongest absorbance at ultraviolet range (BSA at 278 nm). The occurrence of electron transfer of BSA under 405 nm light was further confirmed by a photocurrent responses experiment (new Supplementary Fig. 18a, please refer to the response to comment 2 of reviewer #1). Meanwhile, we presented a new Supplementary Fig. 15b, indicated BSA (0.315 mM) at polymerization concentration might have weak absorbance at 405 nm.

New Supplementary Fig. 15. (b) The UV-Vis absorption spectra of the different concentration of BSA.

Comment 2: It seems this polymerization suffers relatively low efficiency because a high concentration of protein is needed (0.315 mM BSA vs 10% DMA). A rough calculation indicates that the mass ratio of BSA/monomer is ~16% (based on the data in Figure S1 and taking the M_n of BSA 66 kg/mol), which suggests a fair large amount of protein is needed to enable a higher monomer conversion (83%) and the resulting polymer is contaminated with a large amount of protein. This is also problematic for GPC measurements of the resulting polymers, which means efforts should be taken to effectively remove BSA prior to GPC measurements. However, it seemed no such separation effort was taken after the polymerization.

Response: Many thanks for the reviewer's insightful comment. Since the 405 nm LED light as well as the weak light intensity were chosen to ensure high biocompatibility, it thus resulted in the high concentration of protein needed in the system to realize a relative high monomer conversion with limited

time. Moreover, given that the presence of high concentration of proteins in various protein-containing biological media including yeast extract (YE), yeast extract peptone dextrose medium (YPD) as well as living cells (such as average amounts of proteins in yeast cells, 29-65% on a dry weight), we carefully chosen 0.315 mM of BSA in this studied system for the whole study. Alternatively, as shown in Supplementary Table 4 and new Supplementary Table 5, high monomer conversion could be achieved by increasing the light intensity with reduced amount of proteins.

To purify the polymers for GPC measurements, according to the properties of polymers, two general procedures are used.

(1) After polymerization, the solution was freeze-dried, and then dissolved in DMF overnight to ensure that the polymer was completely dissolved. The insoluble protein was removed by filtering.

(2) After polymerization, the proteins could be removed based on salting out by using saturated $(\text{NH}_4)_2\text{SO}_4$ solution to realize the separation between protein and polymer.

Modification: According to the reviewer's comments, we have added the following descriptions into Supplementary Information, Experimental section page 4 paragraph 1 and page 5 paragraph 1 yellow highlighted parts, which read as follows:

General procedure for the kinetic study of DMA in H_2O

"...Each reaction aliquot was dried by freeze-drying and was dissolved in DMF overnight to ensure that the polymer was completely dissolved, while the protein precipitation was removed by 0.45 μm filter for GPC characterization"

Synthesis of block copolymer PDMA₁-b-PDMA₂

"...Further, by adding saturated $(\text{NH}_4)_2\text{SO}_4$ solution into reaction mixture, proteins and polymers were separated due to salt out. After drying in air, pure PDMA₁ was obtained."

Comment 3: It seems Page 3, it was not clear which polymerization system was used for on/off experiments. Such information should be provided in the text and in the Figure captions. Similarly, the polymerization information for each of the panels shown in Figure 2 should be clearly indication in the figure caption.

Response: Many thanks for pointing them out. We have carefully checked the manuscript to ensure that all of the related polymerization conditions were added into the main text and the Figure captions.

Modification: According to your suggestion, the polymerization condition has been added into the revised manuscript in Fig. 2 caption and in main text on page 2 paragraph 1, page 3 paragraph 1 and page 4 paragraph 1, 2, which read as follows:

"**Fig. 2** Kinetic study of BSA-mediated RAFT photopolymerization. ...Polymerization conditions: [BSA] = 0.315 mM, [M] = 10% w/v, (a-h) [BSA]:[PEG-CTA]:[DMA] = 0.144:1:460, [BSA]:[CTA]:[MEA] = 0.144:1:350, (a, b, c, d, g, h) using DMA monomer and (e, f) using MEA monomer, (i) PDMA₁: [BSA]:[PEG-CTA]:[DMA] = 0.162:1:517, 6 hours, PDMA₁-b-PDMA₂: [BSA]:[PEG-CTA]:[DMA] = 0.769:1:2462, 6 hours, violet LED light (20 W, 405 nm), room temperature, Ar protection."

"...Furthermore, the RAFT heterogeneous photopolymerization of MEA was performed to produce polymer nanoparticles (BSA:CTA:MEA = 0.144:1:350)."

"...To demonstrate so, the polymerization kinetic experiment of BSA-mediated RAFT photopolymerization of DMA (BSA:PEG-CTA:DMA = 0.144:1:460) was performed through alternative sequence of light ("ON" and "OFF")."

"...Meanwhile, taking the first block PDMA₁ as a macromolecular CTA which was prepared using a molar ratio (BSA:PEG-

CTA:DMA = 0.162:1:517) under the 405 nm light in H₂O for 6 hours (monomer conversion ratio, 85.7%), by mixing with BSA and DMA, the block polymer PDMA₁-*b*-PDMA₂ was obtained after photopolymerization for 6 hours (BSA:PDMA₁:DMA = 0.769:1:2462, monomer conversion ratio, 80.1%)."

"...the concentration of BSA in the supernatant solution of BSA-mediated photopolymerization of MEA at different times...(BSA:PEG-CTA:MEA = 0.147:1:470)."

Comment 4: Figure 2d, if a line for theoretical molecular weight is also shown, it would be helpful to see the degree of matching between theoretical molecular weights and GPC measurements.

Response: Many thanks for the instructive comment. We have carefully calculated the theoretical molecular weight using the following equation: $M_n^{\text{theo}} = [M]_0/[CTA]_0 \times M_w^{\text{monomer}} \times \alpha + M_w^{\text{CTA}}$, where $[M]_0$, $[CTA]_0$, M_w^{monomer} , α , and M_w^{CTA} correspond to monomer and chain transfer agent concentration, molar mass of monomer, monomer conversion, and molar mass of chain transfer agent. The plot of the theoretical molecular weight was added into Fig. 2d, which presented almost linear relationship similar to the experimental molecular weight M_n^{exp} . M_n^{exp} took polystyrene as the standard sample of GPC measurement, due to the difference in the structure of PDMA and PS, the two plots did not coincide. However, both M_n^{exp} and M_n^{theo} showed almost analogous linear growth with the increase of polymerization time, indicating steady state propagation of polymer chain showing the living characteristic of the polymerization system.

Modification: The plot of M_n^{theo} has been added into Fig. 2d.

Fig. 2 Kinetic study of BSA-mediated RAFT photopolymerization. ...d Evolution of molecular weight and dispersity of PDMA versus monomer conversion ratio.

Comment 5: Figure 2e, the particles seemed to be flattened on the substrate surface in the SEM picture. This is probably because poly(MEA) has a low glass transition temperature (lower than room temperature, which is probably the temperature at which SEM was taken). Thus, the particle size and morphology measured by SEM does not reflect the actual size and morphology of the particles. What data was used for particle size evolution against time, DLS or SEM? This information should be clearly indicated.

Response: We sincerely thank you for your careful checking and comments. The bar chart in Fig. 2e of diameters statistics of PMEA nanoparticles was obtained by DLS measurement in H₂O, and only the inset in Fig. 2e was the SEM image which represented PMEA nanoparticles after 12.8 hours polymerization.

Modification: We has added the related description into the revised manuscript in Fig. 2 caption and main text on page 2 paragraph 1, which read as follows:

Fig. 2 Kinetic study of BSA-mediated RAFT photopolymerization. ...e The diameter statistics of the formed PMEA nanoparticles was based on DLS measurement in H₂O"

"...The PMEA nanoparticles with different diameters over the range between 16 and 80 nm were synthesized by controlling the polymerization time and the diameter of nanoparticles in H₂O was determined by DLS measurement shown in Fig. 2e,"

Comment 6: Figure 2h, the plot of Mn vs time is not very meaningful as it simply reflects the results of Figure 2g. A more meaningful plot is ln(M₀/M_t) vs time, which shows whether first order can still be obtained or not during on/off process.

Response: Your concern about ON-OFF light experiments is important and instructive. We chose the monomer conversion at 0, 1, 3, 5, 6.5 hours to calculate ln(M₀/M_t) and the kinetic plot of ln([M]₀/[M]_t) versus time was shown in new Fig. 2h. Despite undergoing the ON/OFF process, the polymerization system still displayed pseudo-first-order polymerization kinetics, which revealed characteristics of living polymerization behavior with constant radical concentration.

Modification: According to your professional suggestions, we have added new Fig. 2h and the relevant explanations into the revised manuscript in Fig. 2 caption and main text on page 3 paragraph 1, which read as follows:

New Fig. 2 Kinetic study of BSA-mediated RAFT photopolymerization. ...h Pseudo-first-order kinetic plot of ln([M]₀/[M]_t) versus time in the ON/OFF experiments of BSA-mediated RAFT photopolymerization of DMA in the presence or absence of light.

"...The incremental molecular weight of the polymers and the low molecular weight distribution were observed versus time (Supplementary Fig. 13) and as shown in Fig. 2h that the polymerization system still maintained pseudo-first-order polymerization kinetics displaying characteristics of living polymerization with constant radical concentration despite undergoing "ON"/"OFF" procedure."

Comment 7: In the synthesis of PDMA1-b-PDMA2, the information on the synthesis of PDMA1 (PDMA500) should be provided: what was the monomer conversion and how was the polymer purified? Also, such information should be provided for the chain extension step: what was the monomer conversion of the second step of polymerization.

Response: Many thanks for pointing it out. We have made a complete supplement for the experimental process in the revised manuscript.

Modification: The detailed procedure of the synthesis of the first block PDMA₁ and the block copolymer PDMA₁-*b*-DMA₂ has been added into Method “Synthesis of block copolymer PDMA₁-*b*-PDMA₂” in Supplementary Information and the related description in the main text on page 4 paragraph 1 yellow highlighted part, which read as follows:

Synthesis of block copolymer PDMA₁-*b*-PDMA₂

Typically, the synthesis of the first block PDMA₁: BSA (20.92 mg, 0.315 μmol), DMA (100 mg, 1.0088 mmol) and PEG-CTA (4.75 mg, 2.02 μmol) were dissolved in deionized water (1 mL) in a 5 mL of glass vial sealed with a rubber septa. The reaction mixture was degassed by bubbling argon for 25 minutes. The vial was irradiated for 6 h under violet LED light (20 W, 405 nm) at room temperature. Monomer conversion was 85.7%. Furthermore, by adding saturated (NH₄)₂SO₄ solution into the reaction mixture, proteins and polymers were separated due to salting out. After drying, the pure PDMA₁ was obtained, which was then used as macromolecular chain transfer agents for the synthesis of the block copolymer PDMA₁-*b*-PDMA₂. BSA (20.92 mg, 0.315 μmol), DMA (100 mg, 1.0088 mmol) and PDMA₁ (38 mg, 0.82 μmol) were dissolved in deionized water (1 mL) in a 5 mL of glass vial sealed with a rubber septa. The reaction mixture was degassed by bubbling argon for 25 minutes. The vial was irradiated for 6 h under violet LED light (20 W, 405 nm) at room temperature. Monomer conversion was 80.1%. Monomer conversion was determined by ¹H NMR spectrum. The product was dried by freeze-drying and dissolved in DMF for GPC characterization to measure number-average molecular weight (*M_n*) and the polydispersity (*M_w/M_n*).

“...Meanwhile, taking the first block PDMA₁ as a macromolecular CTA which was prepared using a molar ratio (BSA:PEG-CTA:DMA = 0.162:1:517) under the 405 nm light in H₂O for 6 hours (monomer conversion ratio, 85.7%), by mixing with BSA and DMA, the block polymer PDMA₁-*b*-PDMA₂ was obtained after photopolymerization for 6 hours (BSA:PDMA₁:DMA = 0.384:1:1230, monomer conversion ratio, 80.1%).”

Comment 8: The fluorescence of BSA was quenched by CTA and monomers at different efficiency. Could the trend of the different quenching ability of monomers be reflected by polymerization rates? That is, could higher polymerization rate be obtained for monomers with a higher quenching ability, especially for monomers with similar polymerization rate coefficient?

Response: We carefully considered this comment and have added the following experiment to explore the relationship between the polymerization rate and quenching rate. We chose two monomers methyl ethyl methacrylate (EMA) and butyl methacrylate (IEMA) with similar propagation rate coefficient (6.7 and 6.8 °C/10⁻² L mol⁻¹ s⁻¹, respectively) (*Macromolecules* 1997, **30**, 3775-3780), and fluorescence quenching studies of BSA with varying concentrations of EMA or IEMA as quencher were implemented in H₂O: ethanol (1: 1). By Stern-Volmer plots of the ratio *I*₀/*I* (ratio of initial and quenched fluorescence intensity of BSA) versus quencher concentration, the quencher rate coefficient (*K_q*) of IEMA was larger than that of EMA, which were estimated to 0.2885*10¹⁰ and 0.1277*10¹⁰ L mol⁻¹ s⁻¹, respectively (new Supplementary Fig. 22). Then the RAFT photopolymerization was carried out in the presence of BSA, PEG-CTA and monomer (BSA: PEG-CTA: monomer = 0.339:1:493, molar ratio). The kinetics of polymerization of IEMA and EMA were assessed by ¹H NMR spectrum, and the plot of ln([M]₀/[M]_t) of two monomers revealed similar linear relationship versus time, in which IEMA with a higher quencher rate coefficient exhibited a faster polymerization rate compared with that of EMA with a low quencher rate coefficient, indicating the faster electron transfer between BSA and IEMA.

Modification: We have added above information into the revised manuscript including main text on

page 5 paragraph 2, new Supplementary Fig. 22 and Supplementary Table 7, which read as follows:

New Supplementary Fig. 22. (a) Fluorescence quenching studies of BSA quenched with EMA and IEMA in H₂O. Stern-Volmer plots of the ratio I_0/I versus quencher concentration, EMA and IEMA. I_0 and I correspond to the emission intensity in the absence and presence of quencher, respectively. (b) Pseudo-first-order kinetic plot of $\ln([M]_0/[M_t])$ versus time of EMA and IEMA. Data are presented as mean \pm s.d., error bars indicate standard deviations ($n = 3$).

Supplementary Table 7. The parameters of Stern-Volmer plots of BSA quenched by chain-transfer agent (PEG-CTA) or various monomers (DMA, MEA, AML, NIPAM, EMA, IEMA), respectively.^[a]

	PEG-CTA	DMA	MEA	AML	NIPAM	EMA	IBMA
$K_q \cdot 10^{-10}$ (L mol ⁻¹ s ⁻¹)	65.3967	0.971	0.23	1.4142	0.1499	0.1277	0.2885
R^2	0.9998	0.9993	0.9997	0.9978	0.9988	0.9896	0.9996

“...The parameters of Stern-Volmer equation were summarized in Supplementary Table 7, which showed PEG-CTA had the maximum quencher rate coefficient (K_q , $6.53967 \cdot 10^{11}$ L mol⁻¹ s⁻¹) compared with that of the monomers, further indicating the priority of electron transfer between BSA and PEG-CTA. To further demonstrate the relationship between quenching ability and polymerization rate, butyl methacrylate (IEMA) and methyl ethyl methacrylate (EMA) with similar propagation rate coefficient (6.8 and 6.7 °C/10⁻² L mol⁻¹ s⁻¹, respectively)⁶⁰ were chosen. By quenching experiment, IEMA exhibited the larger quencher rate coefficient than that of EMA ($0.2885 \cdot 10^{10}$ and $0.1277 \cdot 10^{10}$ L mol⁻¹ s⁻¹, respectively). The RAFT photopolymerization of two monomers were carried out (BSA:PEG-CTA:monomer = 0.339:1:493, molar ratio), and both the plot of $\ln([M]_0/[M_t])$ of two monomers revealed similar linear relationship versus time while IEMA presented a faster polymerization rates compared with that of EMA, which also indicated the faster quenching rate could correspond to higher polymerization rate (Supplementary Fig. 22).”

Comment 9: The thiol product of CTA in Figure 3e need to be demonstrated in detail, as well as how the CTA maintains the controlled polymer synthesis despite the decomposition into thiols. It seems there is no pathway to regenerate the original tyrosine structure.

Response:

(1) About the thiol product of CTA in Figure 3e.

According to the typical RAFT polymerization procedures (*Macromolecules* 1998, **31**, 5559), the thiol product of CTA in new Fig. 3e was given in detail. Through UV-vis absorption of CTA before and after polymerization, the end group fidelity of CTA was approximately >96%, and the ON/OFF experiment should also indicate high end group fidelity of CTA.

(2) About the regeneration of the original tyrosine structure.

To respond the reviewer's insightful concern, we have detected the amounts of Tyr residues on BSA after polymerization as shown in new Supplementary Fig. 18b. As the reviewer's anticipation, due to the oxidation of Tyr, a clear decrease in the Tyr number on BSA was observed after 9 hours polymerization (please refer to more detailed descriptions in the response to comment 2 of reviewer #1).

Modification: We have improved Fig. 3e and added the above information into Fig. 3e caption, which read as follows:

Fig. 3 Investigation of BSA-mediated photopolymerization mechanism. ...e Proposed mechanism of BSA-mediated controlled radical photopolymerization. Excited-state BSA under light irradiation could enhance its reduction ability significantly, and then excited-state Tyr residues in BSA interacted with monomers or CTA, through proton-coupled electron transfer resulting in the formation of primary radical, which subsequently initiated chain propagation or RAFT process.

Comment 10: In the PTIO experiments, it is helpful to present the UV-vis spectrum of PTIO and its reaction in the presence of BSA. The authors described this type of reaction as an electron transfer process. As I understand, PTIO is a radical trap, which captures a generated radical to form an adduct. Then, the question is: what is identity of the radical generated? EPR experiments would be helpful to identify the generated radical. To clarify this point, either suitable references should be provided on electron transfer to PTIO or EPR should be taken.

Response: We sincerely appreciated your comments and suggestions. We have added the UV-Vis absorption spectrum of PTIO and the mixture PTIO/BSA under 405 nm LED light in H_2O as shown in new Supplementary Fig. 24a, and the maximum absorption spectrum of PTIO was shown at 577 nm, showing a clearly decrease. About PTIO as radical scavenging, the reference “2-Phenyl-4,4,5,5-tetramethylimidazoline-1-oxyl 3-Oxide (PTIO•) Radical Scavenging: A New and Simple Antioxidant Assay In Vitro” has been cited in main text as reference 61 (*J. Agr. Food Chem* 2017, 65, 6288-6297). In order to further confirm the radical generation. Electron spin resonance (ESR) experiment using 5,5-dimethyl-1-pyrroline N-oxide (DMPO) as the spin trapping was carried out in the solution containing BSA and DMA in H_2O (BSA:DMA:DMPO = 0.3:1000:100). After photopolymerization for 3 h under 405 nm light, the reaction solution was quickly transferred to a capillary tube for ESR measurement as

shown in new Supplementary Fig. 23. The x-band ESR spectra clearly confirmed generation of radicals with coupling constants of $\alpha_N = 15.9$ G and $\alpha_{\beta-H} = 21.1$ G.

Modification: We have added the UV-Vis absorption spectrum of PTIO and the mixture PTIO/BSA under 405 nm LED light into new Supplementary Fig. 24a, and the reference about PTIO radical has been added into the main text as reference 61. ESR spectra, experimental process and detailed description have been added into new Supplementary Fig. 23, Method section and main text on page 5 paragraph 3, respectively, which read as follows:

Supplementary Fig. 24. (a) The UV-Vis absorption spectrum of PTIO and the mixture PTIO/BSA under 405 nm LED light in H₂O. [PTIO] = 0.129 mM, [BSA] = 27 μ M.

New Supplementary Fig. 23. ESR spectrum of BSA mediated photopolymerization of DMA. [DMPO] = 100 mM, [DMA] = 1 M, [BSA] = 0.3 mM, violet LED light (20 W, 405 nm), room temperature, 3 h, Ar protection.

In the main text:

"...Moreover, electron spin resonance (ESR) experiment was also employed to confirm the formation of radicals. 5,5-dimethyl-1-pyrroline N-oxide (DMPO) as the spin trapping was mixed with BSA and DMA in H₂O (BSA:DMA:DMPO = 0.3:1000:100). The signal of a relatively stable adduct radical ($\alpha_N = 15.9$ G, $\alpha_{\beta-H} = 21.1$ G) in the ESR spectra clearly revealed radical generation during the photopolymerization (Supplementary Fig. 23)."

In the section of Characterization Methods:

"...Electron spin resonance (ESR) measurement was performed using Bruker EMX PLUS. General instrument parameters were as follows: center field, 3500 G, field scan range, 100 G, microwave power, 6.325 mW, sweep time, 30 s, time constant, 0.01 ms, modulation frequency, 100 kHz, modulation amplitude, 1 G."

General procedure of ESR experiment

In brief, a glass vial (5 mL) charged with BSA (20 mg, 0.301 μ mol), DMA (100 mg, 1.009 mmol), and DMPO aqueous (1 mL, 100 mM) at a molar ratio of [BSA]:[DMA]:[DMPO] = 0.3:1000:100 was sealed with a rubber septa. The reaction mixture was degassed by bubbling argon for 25 minutes. The vial was irradiated under violet LED light (20 W, 405 nm) at room temperature for 3 h, and then the reaction solution was quickly transferred to a capillary tube for ESR measurement.

Comment 11: The authors showed that the 3D structure of protein was not necessary for the polymerization because aggregated and denatured BSA could also initiate the polymerization. Then, the question is why protein was used for the polymerization (a lot of mass is wasted), not simply tyramine (the amine group needs to be protected in this case to avoid aminolysis of RAFT agent)?

Response: Many thanks for the insightful comments. Our main considerations were based on the following two aspects: (i) In terms of the development of the polymerization method used in intra-organism polymerization, the widely existed proteins was a good initiator choice which then avoid additional incorporating small molecular initiators, since the addition of foreign matters might more or less cause damage to the organism. Based on this concern, we try to investigate the possibility by directly employing proteins which are abundant in living organisms to initiate polymerization. (ii) Besides avoiding any possible aminolysis of RAFT agent, the solubility of Tyr is low in H₂O, which is ~about 1.66 mM. It is equivalent to 0.083 mM BSA according to the content of 20 tyrosine residues in each BSA, which limited the implementation and application of Tyr-based polymerization.

Comment 12: Information of the amounts of other proteins or their molar ratios relative to CTA and monomers used in polymerization should be given.

Response: Thank you for pointing them out. We have carefully checked the manuscript to ensure that all polymerization conditions have been provided in detail, involving main text on page 4 paragraph 2, Fig. 3 caption, Supplementary Fig. 9, 10, 12, 13, 25 and Supplementary Table 4, which read as follows.

"Based on bradford assay, the concentration of BSA in the supernatant solution of BSA-mediated photopolymerization of MEA at different times was determined...(BSA:PEG-CTA:MEA = 0.147:1:470)."

"Supplementary Fig. 9. ¹H NMR spectrum of BSA-mediated RAFT photopolymerization of DMA at different points in D₂O. ...[BSA] = 0.315 mM, [M] = 10% w/v, [BSA]:[PEG-CTA]:[DMA] = 0.144:1:460, violet LED light (20 W, 405 nm), room temperature, Ar protection."

"Supplementary Table 4. RAFT photopolymerization of MEA mediated by the different concentrations of BSA. ...[PEG-CTA] = 1.708 mM, [MEA] = 10% w/v, violet LED light (20 W, 405 nm), room temperature, Ar protection."

"Supplementary Fig. 10. The study of RAFT photopolymerization of MEA mediated by different concentrations of BSA. ...[PEG-CTA] = 1.708 mM, [MEA] = 10% w/v, violet LED light (20 W, 405 nm), room temperature, Ar protection."

"Supplementary Fig. 12. The GPC traces of BSA-mediated RAFT photopolymerization of (a) NIPAM, (b) AML and (c) PEGA, respectively. [BSA] = 0.315 mM, [PEG-CTA] = 1.07 mM, [M] = 10% w/v, violet LED light (20 W, 405 nm), room temperature, Ar protection."

"Supplementary Fig. 13. GPC traces of PDMA in the ON/OFF experiments at different time intervals. [BSA] = 0.315 mM, [M] = 10% w/v, [BSA]:[PEG-CTA]:[DMA] = 0.144:1:460, violet LED light (20 W, 405 nm), room temperature, Ar protection."

"Supplementary Fig. 25. The GPC traces of RAFT photopolymerization of MEA, mediated by (a) GOx, (b) Lysozyme and (c) RNase-A, respectively. The insets were the photographs of polymerization at 0 h and 6 h. [GOx] = 0.048 mM, [Lysozyme] = 0.505 mM, [RNase-A] = 0.608 mM, [PEG-CTA] = 1.824 mM, [M] = 10% w/v, violet LED light (20 W, 405 nm), room temperature, Ar protection."

"**Fig. 3** Investigation of BSA-mediated photopolymerization mechanism. ...Molecular weight (M_n) and polydispersity (M_w/M_n) determined by GPC analysis (DMF used as eluent). Conversion ratio was determined by ^1H NMR. Polymerization conditions: [M] = 10% w/v, [BSA]/[CTA]/[MEA] = 0.147:1:470, [HSA] = 0.125 mM, [GOx] = 0.048 mM, [Lysozyme] = 0.505 mM, [RNase-A] = 0.608 mM, [PEG-CTA] = 1.824 mM, violet LED light (20 W, 405 nm), room temperature, Ar protection."

Comment 13: Figure S22f, what were the molecular weights and dispersity of the polymers.

Response: We have added the molecular weights and dispersity of the polymers into the Supplementary Fig. 29f in the revised Supplementary Information.

Supplementary Fig. 29. ... (f) The GPC curves after polymerization for 1.5 h.

Comment 14: It is not surprising that photopolymerization proceeded under aerobic conditions when high concentrations of proteins were used, since some of the generated radicals consume oxygen. However, it is important to evaluate how the polymerization was affected by the presence of air. Are there any induction periods observed due to initial consumption of oxygen? How are the quality of the obtained polymers affected by the presence of oxygen and the excess use of protein?

Response: Thank you for the thoughtful question. The aerobic photopolymerizations of MEA and DMA mediated by BSA were implemented by using 1.2 mM of BSA and monomer (10% w/v), respectively. By ^1H NMR spectrum, the kinetics plot of $\ln([M]_0/[M]_t)$ of DMA revealed the induction periods at ~2 h, and the turbidity of polymerization of MEA showed the induction periods at 1~2 h shown in new

Supplementary Fig. 30b, c. We can estimate that the oxygen content is about 13.98 μmol in 2 mL of vial containing 0.5 mL of solution according to the oxygen content in the air 21%, and the rate of oxygen consumption was estimated to be 0.155-0.117 $\mu\text{mol min}^{-1}$. After the aerobic photopolymerization, the successful synthesis of PDMA was confirmed by ^1H NMR spectrum, exhibiting a spectrum similar to that of PDMA synthesized under anaerobic conditions (Supplementary Fig. 30d).

Modification: We have made relevant modifications into the revised Supplementary Information by adding new Supplementary Fig. 30 and the caption, which read as follows:

New Supplementary Fig 30. The investigation of aerobic photopolymerization. (a) Photograph of BSA-mediated photopolymerization of MEA, $[\text{MEA}] = 10\%$ w/v. (b) Photograph of BSA-mediated photopolymerization of MEA versus time, and (c) the kinetic plot of $\ln([M]_0/[M]_t)$ versus time of DMA, 1.2 mM BSA, which revealed the induction periods of the polymerization at 1.5-2 h and the rate of oxygen consumption was estimated to be 0.155-0.117 $\mu\text{mol min}^{-1}$. (d) ^1H NMR spectrum of PDMA in D_2O . $[\text{M}] = 10\%$ w/v, violet LED light (20 W, 405 nm), aerobic condition.

Comment 15: I donot understand why in the absence of light highly fluorescent was still observed in Figure 5d.

Response: Fig. 5d exhibited the viability of yeast cells after being treated and untreated with 405 nm light for 40 min, and both yeast cells were stained by FDA (green) and PI (red). Both yeast cells showed highly green fluorescent from FDA staining after 134 hours incubation, indicating the cells maintained normal proliferation and 95% cell viability were observed.

Modification: In order to avoid any confusion, we have added related description in revised manuscript in main text on page 8 paragraph 1, which reads as follows:

"...both cells treated and untreated with 405 nm LED light showed highly green fluorescent from FDA, indicating that the yeast cells still maintained normal proliferation, and more than 95% cell viability was observed after 134 hours."

Comment 16: Photos of photopolymerization experimental setup can be provided in the supporting information.

Response: Many thanks for the suggestion. We have added photograph of photopolymerization experimental setup into Supplementary Information as new Supplementary Fig. 1.

New Supplementary Fig. 1. The Photograph of photopolymerization experimental setup.

Response to the reviewers' comments (Reviewer#3)

Comments to the Author:

The presented paper deals with photo-induced polymerization using different Tyr-containing proteins as polymerization initiators (called by authors as a potential new biocompatible polymerization system). This is an interesting idea that could be used further in research works. The work seems to be complete and authors applied plethora of analytical methods including kinetic analysis and cell imaging. Experiments appear to be carried out professionally, but I have a feeling, that the structure of presented results is messy and overloaded with collected data, thus it is difficult to follow.

Response: Many thanks for the reviewer's agreement on our work and the helpful comments. We have carefully checked all the manuscript and supplementary materials, including the grammar, expression, and some supplement of details and experiments related with mechanism to improve logicity and coherence of the text. All of the modifications have been highlighted with yellow colour in the revised manuscript and Supplementary Information. The main modifications are listed as follows:

- (i) Add detailed description of polymerization including composition, concentration and polymerization conditions, and add detailed explanations in the manuscript, involving Fig. 2 caption, Fig. 3 caption, Fig. 5d, e, Supplementary Fig. 9, 10, 12, 13, 25 and Supplementary Table 4 (shown yellow highlighted parts in the revised manuscript).
- (ii) Delete some wordy expressions including main text and captions.

However, in my opinion, I have doubts, if the data interpretation and proposed explanation are correct. In my knowledge, and also as presented in UV-Vis spectra in Supplementary data, tyrosine and proteins containing Tyr would not absorb light, especially to form the excited singlet state, at 405 nm. In fact, I would feel more convinced, if authors can present the UV vis spectra of the mixture protein/monomers applied in experiments together with separated UV-Vis for monomers and proteins to check, if there is any interactions before (ground state complex?). Other wise, authors should think again about the possible explanation of polymerization occurring in this system.

Response: We really appreciated the reviewer's insightful comments. As this question was also raised by Reviewer #2 in comment 1, we include below a repeat of our response:

- (i) In this study, we aim to search for a mild and biocompatible polymerization system which should be

the key point to realize intrinsically non-natural macromolecules synthesis for modulating living organism function/behaviour. Accordingly, to keep the biocompatibility of the polymerization system at most, a 405 nm LED light with a weak light intensity (1.38 mW/cm^2) is chosen finally with compromised polymerization rate despite proteins have the strongest absorbance at ultraviolet range (BSA at 278 nm). The occurrence of electron transfer of BSA under 405 nm light was further confirmed by a photocurrent responses experiment (new Supplementary Fig. 18a, please refer to the response to comment 2 of reviewer #1).

(ii) According to the reviewer's suggestion, we also performed the two experiments about UV-vis spectra of the mixture protein/monomers applied in experiments together with separated UV-Vis for monomers and proteins. Firstly, the polymerization of different concentrations of BSA and MEA (60 mg/mL) was employed, and the UV absorbance of the solution before polymerization, the supernatant after polymerization for 14 h (the polymer was removed by centrifugation), BSA and MEA (60 mg/mL) were checked. By comparison, no new UV absorption peaks were found, and the UV absorption curves of the mixture solution before polymerization were similar to theoretical curves calculated by adding UV absorption of the single MEA and BSA, as shown in new Supplementary Fig. 21a. Further, the UV-vis absorption of BSA ($A_{\text{max}} = 278 \text{ nm}$), DMA, PEG-CTA ($A_{\text{max}} = 308 \text{ nm}$) and mixture BSA+DMA, BSA+PEG-CTA and BSA+DMA+PEG-CTA applied in experimental concentration were detected, showing no new absorption peaks (new Supplementary Fig. 21b). Based on the above analysis, it should indicate the monomer and protein do not interact with each other in the form of ground-state complexes before polymerization.

Modification: Besides the modifications in the response to the second reviewer comment 1, the UV-Vis absorption spectrum of mixture BSA, MEA, DMA and PEG-CTA together with separated UV-Vis for monomers and proteins have been added into new Supplementary Fig. 21, which reads as follows:

New Supplementary Fig. 21. (a) The UV-Vis absorption spectrum of the mixture BSA/MEA solution before polymerization, the supernatant after polymerization for 14 h, separated BSA and MEA. (b) The UV-Vis absorption spectrum of BSA, PEG-CTA and DMA together and separated UV-Vis. [BSA] = 0.274 mM, [DMA] = 10% w/v, [PEG-CTA] = 1.163 mM.

Reviewers' Comments:

Reviewer #2:

Remarks to the Author:

The authors have done an excellent revision and I fully support its publication and suggest acceptance.

The following point may be considered during the post-acceptance period:

In Supplementary Table 4, it is strange that for monomer conversions in the range of 27-72%, the Mn values are essentially the same. Similar issues occur In Supplementary Fig 11 and Supplementary Table 5: it is strange that the Mn is all around 100 kg/mol (GPC traces have almost the same retention time) at different monomer conversions, ranging from 53.7 to 91.8%.

Reviewer #3:

Remarks to the Author:

The corrected manuscript gives now large studies carried out with good level of competence using appropriate methods. Authors introduced corrections and answered questions explaining carefully their results. I don't have more comments nor questions.

Response to the reviewers' comments (Reviewer#2)

Comments to the Author:

The authors have done an excellent revision and I fully support its publication and suggest acceptance. The following point may be considered during the post-acceptance period: In Supplementary Table 4, it is strange that for monomer conversions in the range of 27-72%, the M_n values are essentially the same. Similar issues occur In Supplementary Fig 11 and Supplementary Table 5: it is strange that the M_n is all around 100 kg/mol (GPC traces have almost the same retention time) at different monomer conversions, ranging from 53.7 to 91.8%.

Response: We sincerely appreciate the reviewer's professional guidance and positive recommendation to our manuscript for publication. Many thanks for the careful checking.

We have carefully considered integrating the complete curve area again to recalculate the molecular weight and its distribution.

(1) When the concentration of BSA was 0.025 mM, the controllability of polymerization was poor with wide molecular weight distribution, showing an obvious asymmetric and trailing GPC curves in Supplementary Fig. 10b. The integral interval was selected from 11 minute to 18.5 minute, and a new annotation has been added to the legend in Supplementary Table 4 as [b].

(2) We presented new normalized GPC curves in Supplementary Fig 11, which more clearly showed that the molecular weight increased with the increase of monomer conversion, and the applicable intervals were integrated to recalculate the molecular weight and its distribution in Supplementary Table 5.

Response to the reviewers' comments (Reviewer#3)

Comments to the Author:

The corrected manuscript gives now large studies carried out with good level of competence using appropriate methods. Authors introduced corrections and answered questions explaining carefully their results. I don't have more comments nor questions.

Response: We sincerely thank the reviewer for the positive evaluation and professional comments to improve the quality of this manuscript.